# Revisiting Active Learning in the Era of Vision Foundation Models

**Sanket Rajan Gupte**[*]                                    *sanketg@stanford.edu*
*Department of Computer Science*
*Stanford University*

**Josiah Aklilu**[*]                                          *josaklil@stanford.edu*
*Department of Biomedical Data Science*
*Stanford University*

**Jeffrey J. Nirschl**                                       *jnirschl@stanford.edu*
*Department of Pathology*
*Stanford University*

**Serena Yeung-Levy**                                        *syyeung@stanford.edu*
*Department of Biomedical Data Science*
*Stanford University*

**Reviewed on OpenReview:** *https://openreview.net/forum?id=u8K83M9mbG*

## Abstract

Foundation vision or vision-language models are trained on large unlabeled or noisy data and learn robust representations that can achieve impressive zero- or few-shot performance on diverse tasks. Given these properties, they are a natural fit for *active learning* (AL), which aims to maximize labeling efficiency. However, the full potential of foundation models has not been explored in the context of AL, specifically in the low-budget regime. In this work, we evaluate how foundation models influence three critical components of effective AL, namely, 1) initial labeled pool selection, 2) ensuring diverse sampling, and 3) the trade-off between representative and uncertainty sampling. We systematically study how the robust representations of foundation models (DINOv2, OpenCLIP) challenge existing findings in active learning. Our observations inform the principled construction of a new simple and elegant AL strategy that balances uncertainty estimated via dropout with sample diversity. We extensively test our strategy on many challenging image classification benchmarks, including natural images as well as out-of-domain biomedical images that are relatively understudied in the AL literature. We also provide a highly performant and efficient implementation of modern AL strategies (including our method) at `https://github.com/sanketx/AL-foundation-models`.

## 1 Introduction

Foundation models (Oquab et al., 2023; Cherti et al., 2022; Bommasani et al., 2022) have become ubiquitous in computer vision. The advent of large vision and vision-language models, pretrained on web-scale data corpora, has led to the significant enhancement of visual representation spaces. Despite differences in pretraining (e.g. contrastive frameworks with language supervision as in Cherti et al. (2022) or vision-only instance discrimination objectives as in Oquab et al. (2023)), the shared theme among vision foundation models are robust learned representations that can be effectively leveraged for state-of-the-art performance

---

[*]Equal contribution.

on multiple downstream tasks. These learned visual features have direct implications for machine learning paradigms designed to mitigate data paucity challenges, most notably deep active learning.

Active learning (AL) (Sener & Savarese, 2018; Ash et al., 2020; Gal et al., 2017; Kirsch et al., 2022; Aklilu & Yeung, 2022; Hacohen et al., 2022; Yehuda et al., 2022; Parvaneh et al., 2022; Mahmood et al., 2022) is a machine learning framework that addresses the scarcity of labeled data within a limited label budget. It achieves this by iteratively requesting labels from an external oracle, such as a subject matter expert or clinician. Stated differently, the challenge is determining the most beneficial labels to query when constrained by a limited labeling budget to maximize model performance. This approach is especially pertinent in practical computer vision applications that necessitate the creation of domain-specific datasets for custom tasks. A well-designed AL strategy can optimize resource allocation and substantially lower acquisition costs in situations where labeled data are scarce and labor-intensive to produce — such as annotating high-resolution histopathology images for tumor classification.

Previous research has tested the benefit of using pretrained representations during AL by leveraging representations of unlabeled data to 1) address the cold-start problem by selecting a good initial pool of candidates for labeling Hacohen et al. (2022); Chen et al. (2023); Chandra et al. (2020), 2) improve query functions by selecting points that maximize the coverage of the representation space Yehuda et al. (2022); Sener & Savarese (2018) and, 3) incorporating unlabeled samples in the training process via semi-supervised learning Gao et al. (2020); Simeoni et al. (2021). However, these studies limit their focus to supervised pretraining on ImageNet or self-supervised pretraining on available unlabeled data, which often proves impractical in many real-world scenarios. For instance, biomedical imaging datasets exhibit distribution shifts from natural images and may not have sufficient samples for effective self-supervised representation learning.

Foundation models offer a compelling solution to these challenges. Image embeddings extracted from these models are semantically organized in the representation space, as evidenced by impressive zero- and few-shot performance with simple linear models, even on fine-grained classification datasets. Evidence also suggests they are robust to domain shifts and generalize well to out-of-distribution data. Since these embeddings "just work" out of the box, building dataset-specific feature extractors is no longer necessary. For these reasons, we envision that future research in AL will build upon the ever-increasing capabilities Dehghani et al. (2023); Sun et al. (2023) of Vision and Vision-Language foundation models to formulate efficient and scalable acquisition strategies. However, there is limited research Tran et al. (2022) on the impact of foundation models on AL. Our work seeks to re-examine the pillars of effective AL strategies in this context, specifically in the low-budget regime in which only a handful of images per class can be annotated. We hope our study's compelling results and novel insights will spur the use of foundation models as a "batteries included" launchpad for future AL research.

**Our key contributions are the following:**

1. We investigate the impact of large-scale vision foundation models on the four pillars of active learning strategies, specifically in the low-budget regime, and contrast our findings with established results. Our analysis systematically explores the following dimensions of a successful AL strategy:

   - We study the impact of initial pool selection and highlight differences with existing research on the cold-start problem.
   - We explore the importance of querying diverse samples and demonstrate that with a few simple modifications, poorly performing AL query strategies can match those that explicitly incorporate sample diversity.
   - We compare representative sampling with uncertainty-based sampling and counter the existing notion that uncertainty-based query methods are ineffective in the low-budget regime.
   - We demonstrate the difficulties of leveraging semi-supervised learning as a complementary approach for building accurate models with limited labeled data.

2. As a direct consequence of the results of our investigation, we construct a simple, performant, and scalable AL strategy, **DropQuery** . This method, built atop the rich semantic features generated by powerful foundation models, leverages an intelligent strategy for initial pool selection, utilizes an

uncertainty-based criterion to generate a pool of query candidates, and selects a diverse subset of candidates for annotation. Our strategy outperforms the current state-of-the-art on a diverse collection of datasets, including fine-grained image classification, out-of-distribution biomedical images, and image classification at scale, all of which are relatively understudied in AL.

## 2 Background and Related Works

Traditional research in AL for image classification has largely focused on the development of acquisition functions to query samples for labeling by an oracle. These strategies have spanned uncertainty-based sampling approaches (Shannon, 1948), Bayesian methods (Gal et al., 2017; Kirsch et al., 2022), strategies to maximize sample diversity, query by committee strategies (Melville & Mooney, 2004), and techniques to estimate the largest update to the model parameters (Ash et al., 2020).

Uncertainty-based methods supply a single measure of uncertainty (epistemic or aleatoric) for unlabeled samples so as to rank them for acquisition. **Entropy** selects instances with maximum predictive entropy, **Uncertainty** selects instances with the lowest confidence, and **Margins** selects instances with the lowest margin between the highest and second-highest predicted class probabilities. The Bayesian approaches to AL (Gal et al., 2017; Kirsch et al., 2019; Woo, 2023) leverage Bayesian neural networks to model uncertainty and the potential informativeness of unlabeled instances. Bayesian Active Learning by Disagreement or **BALD** (Gal et al., 2017) aims to query instances that maximize the mutual information gain between model parameters and predictive outputs. Additionally, the **PowerBALD** (Kirsch et al., 2022) query challenges the traditional top-$B$ acquisition approach by noting the correlation of queried instances *between* each iteration of AL, which is alleviated by introducing stochasticity in the query.

The prior works that have placed an emphasis on diversity sampling argue that the classifier should learn good decision boundaries in the representation space early. In the lens of these diversity-based queries, the AL paradigm can be re-framed as *"what are the diverse instances that are representative of the data distribution?"*. The **Coreset** algorithm (Sener & Savarese, 2018) can be approximated as solving the $k$-Center greedy problem (Farahani & Hekmatfar, 2009), where the distance between any instance and its nearest cluster center (the labeled instances from previous AL iterations) is minimized. Following the Coreset query, the recently developed **ProbCover** query aims to avoid the selection of outlier instances and query more representative points. This is done by recasting the AL problem as a max coverage problem (Nemhauser et al., 1978).

A hybrid technique, **BADGE** (Ash et al., 2020), uses an AL query that inspects the gradients of a neural network's final layer relative to the current model's prediction. This identifies unlabeled instances that could induce significant model updates. The underlying intuition is that samples prompting large gradient updates require substantial adjustments, signifying uncertainty in samples with new features. While this method yields diverse and highly uncertain samples, it is computationally intensive due to the storage of penultimate activations. **Alfa-Mix** (Parvaneh et al., 2022) solicits unlabeled samples with the greatest classification prediction variability when their representations are linearly interpolated with labeled instances, suggesting that the model needs to learn new features from unlabeled data points. **Typiclust** (Hacohen et al., 2022), queries typical instances during the early AL stages leveraging the representation space. Clustering in a semantically meaningful feature space after self-supervised learning ensures maximum diversity sampling. Sampling dense regions within clusters aids in better classifier learning within feature space.

However, in most of the settings considered by these works, a model is trained from scratch at each iteration using a random initialization or an ImageNet pretrained backbone. The resulting models are often poorly calibrated (Guo et al., 2017) and fail to provide robust estimates of uncertainty. These settings also require a relatively large labeling budget as training deep non-linear models with limited labels can be challenging (Yuan et al., 2020). Consequently, a number of such strategies have been shown to underperform random sampling in the low budget (Simeoni et al., 2021; Hacohen et al., 2022), or do not perform significantly better than random when strong regularization is applied (Munjal et al., 2022).

Recent methods designed for the low budget regime (Hacohen et al., 2022; Yehuda et al., 2022) take a more holistic approach to developing an effective AL strategy by leveraging pretrained model embeddings to cluster

features and select representative points, which has been shown to improve the selection of the initial labeled pool. Other approaches aim to exploit unlabeled points using self-supervised or semi-supervised learning, or a combination of both (Chan et al., 2020; Bai et al., 2021). Their findings demonstrate that leveraging rich representations significantly improves AL performance in the context of a small labeling budget (Lüth et al., 2023).

Given these promising results, it is reasonable to conjecture that embeddings from large-scale vision transformers trained on billions of images would amplify the efficacy of these complementary components, constructing an effective strategy that surpasses a simpler baseline consisting of only a standalone acquisition function. To comprehensively investigate these facets of AL, we compare the impact of a simple pool initialization strategy that leverages the representation space's implicit structure to sample diverse and representative points. Further, we investigate different classes of query functions in this context to compare uncertainty-based approaches with typicality-based methods, as the latter have been shown to significantly outperform the former in the low-budget regime. Orthogonal to the query function, we investigate if semi-supervision via label propagation can be an effective aid in improving the performance of an active learner.

Although some recent work has explored the effects of pretraining on the AL query (Hacohen et al., 2022; Tamkin et al., 2022), there has not been a rigorous evaluation of AL queries with respect to developments in foundation models and vision-language pretraining that have yielded highly expressive and rich visual representation spaces. In the following sections, we outline the critical components necessary for an effective AL query in the context of large foundation models.

## 3 Investigating the pillars of effective active learning strategies

Formally, we consider the batch AL setting: let $\mathcal{D} = \{\mathcal{D}_{\text{pool}} \cup \mathcal{D}_{\text{test}}\}$ be a dataset with a held-out test set and training pool $\mathcal{D}_{\text{pool}} = \{\mathcal{D}_U \cup \mathcal{D}_L\}$ consisting of unlabeled instances $\{\boldsymbol{x}_i\}$ for $i \in \{N_U\}$ and labeled instances $\{(\boldsymbol{x}_j, y_j)\}$ for $j \in \{N_L\}$. Let $f : \mathbb{R}^{H \times W \times 3} \to \mathbb{R}^d$ be a feature extractor such that $\boldsymbol{z}_i := f(\boldsymbol{x}_i) \in \mathbb{R}^d$ is a feature vector and $d$ is the embedding dimension. Let $g(\boldsymbol{z}; \boldsymbol{\theta}) : \mathbb{R}^d \to \mathbb{R}^K$ be a classifier parameterized by $\boldsymbol{\theta}$. The typical AL procedure consists of a series of sequential iterations beginning at $t = 0$ where the labeled pool $\mathcal{D}_L$ is the empty set. At each iteration, a querying function $a(\{\boldsymbol{z}_i : i \in \{N_U\}\}; \boldsymbol{\theta}_t)$, which given the current model parameters $\boldsymbol{\theta}_t$, receives as input all unlabeled instances, selects $B$ instances to be labeled by an external oracle (e.g. clinical expert). Once labels are acquired, these labeled instances are subsumed in $\mathcal{D}_L$ and removed from the unlabeled pool, and the model is retrained on this slightly larger labeled pool. We simulate the AL procedure by hiding labels $y_i$ until queried for a chosen instance $\boldsymbol{x}_i$.

In this work, we employ frozen vision-only or vision-language foundation models as the feature extractor $f(.)$ and a linear head as the classifier $g(.)$. We only require a single forward pass through the feature extractor to generate embeddings which are saved for use in our various experiments. This enables us to study a wide range of experimental settings for AL efficiently. Unless mentioned otherwise, we use a DINOv2 VIT-g14 (Oquab et al., 2023) transformer as the feature extractor. Simple linear models trained on features extracted by this model achieve performance that is just shy of state-of-the-art, making them a compelling alternative to end-to-end fine-tuning of the backbone with scarce labels. Following the recommendations of (Munjal et al., 2022), we apply strong regularization to our models in the form of weight decay (1e-2) and aggressive dropout (0.75). Note that the benchmarks used to evaluate the various AL methods in this work were not used in the pretraining of the foundation models we investigated (we refer the reader to the data curation processes outlined by Oquab et al. (2023)).

In our experiments, we set a query budget $B$ of 1 sample per class per iteration. For instance, the CIFAR100 dataset contains images corresponding to 100 classes, so the query budget is set to 100 samples. Note that setting $B$ to 1 sample per class per iteration does not necessarily imply that exactly one sample is chosen from each class, since the class labels are unknown at the time of querying. We run our active learning loop for 20 iterations and average our results over 5 random seeds for each experiment. We analyze the following query functions: Random Sampling baseline, Entropy, Uncertainty, Margins, Core-Set (greedy k-centers), BALD, Power BALD, BADGE, ALFA-Mix, Typiclust, and ProbCover. The hyper-parameters for these query functions, if any, are taken from the original publication.

### 3.1 The impact of initial pool selection

We begin our study by measuring the impact of different initial pool selection strategies in AL, particularly in the *cold-start* scenario where no data has been used for training the classifier. It has been shown in the recent literature Hacohen et al. (2022); Yehuda et al. (2022); Chen et al. (2023) that the initial pool is crucial for establishing a good rudimentary classifier that enjoys performance gains throughout AL. Hacohen et al. (2022) and Yehuda et al. (2022) also argue that the AL acquisition function need not be decoupled from the initial pool selection. Rather, an AL query should acquire representative samples so the classifier can learn good decision boundaries from the first query. Intuitively, in order for an AL query to be effective, it must rely on the current classifier's estimates of uncertainty or informativeness.

To assess initialization strategies, we contrast AL query performance when the initial training pool for the active learner is randomly selected for labeling to a centroid-seeking approach following Pourahmadi et al. (2021), where the initial pool consists of samples from $\mathcal{D}_U$ that are the closest to class centers after $K$-means clustering in the feature space, where $B$ is the number of clusters. **Typiclust** and **ProbCover** utilize different initialization strategies based on sample density and maximizing coverage, respectively, so we hold these queries true to their initialization approach. Table 1 compares the deltas ($\Delta$) in the test set performance of a linear classifier actively trained on foundation model features using a randomly initialized initial pool versus a centroid-based initial pool. Note that a random initial pool is suboptimal to methods that explicitly take advantage of semantically meaningful embedding spaces (i.e. **Typiclust** or **ProbCover**). The tremendous performance gains in early AL iterations for uncertainty-based AL queries like **Entropy** sampling enable these methods to surpass queries like **Typiclust** and **ProbCover**.

Table 1 also demonstrates the impact of initial pool selection on AL queries acting on foundation model features. Some queries, like **Uncertainty**, **Entropy**, and **Coreset**, enjoy performance gains throughout AL given a centroid-based initialization, a major boost to the random initialization. Interestingly, some queries like **Alfa-Mix** and **BADGE** have deltas that converge within 2-4 iterations, and in later iterations (8-16) we observe higher accuracy with a randomly selected initial pool. Since the **Alfa-Mix** query crucially relies on interpolation in the feature space, we hypothesize that the separability of foundation model representation space renders differences between initial pooling strategies negligible. After a few iterations, the robust visual representations from foundation models enable the selection of highly representative samples, which help the classifier establish good class boundaries in few-shot.

The results from this experiment are in contrast to previous preliminary findings from Chandra et al. (2020) that report no significant difference in initial pool selection strategies in the long run. While we also see diminishing returns as we progress to later iterations, we emphasize that the experimental results in Chandra et al. (2020) also showed little to no difference caused by most initialization strategies, even in the very first iteration of AL. This observation does not align with our results in which we see stark differences caused by intelligent selection of the initial labeled pool in the beginning iterations.

However, this discrepancy can be resolved when taking into account the sizes of the labeling budgets. Our experimental setting is the very low-budget regime, and the number of labeled samples in the final iterations of our studies barely overlaps with the starting budgets studied in Chandra et al. (2020). We conclude that in the low-budget regime, where only a few examples per class have been acquired, initialization is crucial for AL performance, but in later iterations, as the number of labeled samples grows or as we transition to higher-budget regimes, it is not as significant and a randomly selected initial pool works just as well.

### 3.2 On the importance of diversity in query selection

In line with the initial pool selection, we evaluate how partitioning the representation space via clustering is crucial for AL performance in subsequent iterations, even after the initial query. To conduct these experiments, we allow AL queries that use stand-alone measures of uncertainty (i.e. **Uncertainty**, **Entropy**, **Margins**, and **BALD**) to deviate from the top-$B$ acquisition pattern persistent in the AL literature. As noted by Kirsch et al. (2022), top-$B$ acquisition can potentially lead to a correlation between queried batches for some $t_i$ and $t_j$, which can hurt performance.

Table 1: Effect of initial pool selection on performance. Test set accuracy using our centroid-based initialization. In parentheses, we show the difference ($\Delta$) in performance when utilizing our centroid initialization vs. random initialization, where a positive ($\Delta$) shown in green indicates improvement over random initialization. We show AL iterations $t$ for datasets CIFAR100 (Krizhevsky, 2009), Food101 (Bossard et al., 2014), ImageNet-100 (Gansbeke et al., 2020), and DomainNet-Real (Peng et al., 2019) (from top to bottom) with DINOv2 ViT-g14 as the feature extractor $f$. For both the Typiclust and ProbCover queries, we report the test accuracy using their own respective initialization strategies for the cold start.

| $t$ | Random | Uncertainty | Entropy | Margins | BALD | pBALD | Coreset | BADGE | Alfa-mix | Typiclust | ProbCover |
|---|---|---|---|---|---|---|---|---|---|---|---|
| | | | | | CIFAR100 | | | | | | |
| 1 | 72.4 (+24.4) | 72.4 (+24.4) | 72.4 (+24.4) | 72.4 (+24.4) | 72.4 (+24.4) | 72.4 (+24.4) | 72.4 (+24.4) | 72.4 (+24.4) | 72.4 (+24.4) | 64.6 | 62.3 |
| 2 | 78.1 (+13.6) | 76.7 (+14.0) | 76.5 (+18.4) | 79.1 (+9.2) | 78.6 (+11.3) | 80.3 (+9.0) | 77.7 (+13.2) | 79.8 (+7.3) | 80.7 (+2.7) | 80.8 | 76.2 |
| 4 | 82.5 (+3.7) | 80.6 (+5.9) | 78.8 (+8.4) | 84.0 (+1.4) | 82.1 (+1.7) | 85.2 (+1.3) | 81.9 (+3.7) | 84.6 (+0.5) | 83.7 (-0.1) | 86.8 | 81.9 |
| 8 | 86.4 (+0.4) | 85.3 (+1.1) | 84.0 (+2.6) | 88.3 (+0.4) | 86.3 (+1.1) | 88.8 (+0.4) | 84.5 (+0.2) | 88.8 (-0.1) | 87.3 (-0.5) | 88.4 | 86.5 |
| 16 | 89.2 (-0.0) | 89.2 (-0.0) | 88.6 (+0.8) | 90.9 (+0.2) | 89.0 (+0.6) | 90.9 (+0.1) | 88.0 (-0.3) | 90.7 (-0.2) | 90.4 (-0.4) | 89.3 | 89.1 |
| | | | | | Food101 | | | | | | |
| 1 | 69.6 (+22.3) | 69.6 (+22.3) | 69.6 (+22.3) | 69.6 (+22.3) | 69.6 (+22.3) | 69.6 (+22.3) | 69.6 (+22.3) | 69.6 (+22.3) | 69.6 (+22.3) | 68.3 | 66.1 |
| 2 | 74.5 (+10.1) | 69.6 (+18.8) | 69.4 (+19.5) | 73.0 (+10.3) | 69.7 (+18.9) | 73.7 (+6.6) | 69.9 (+18.5) | 72.9 (+8.2) | 77.1 (+3.0) | 79.1 | 72.3 |
| 4 | 79.3 (+2.1) | 73.2 (+9.7) | 72.1 (+13.5) | 78.1 (+2.7) | 72.1 (+9.9) | 79.0 (+0.8) | 71.7 (+8.7) | 78.7 (+1.2) | 80.5 (+0.9) | 83.1 | 78.1 |
| 8 | 83.9 (+0.1) | 79.4 (+2.2) | 77.9 (+4.9) | 85.0 (+0.3) | 78.0 (+3.7) | 85.4 (-0.3) | 76.8 (+3.5) | 85.5 (+0.2) | 85.1 (-0.5) | 86.0 | 81.9 |
| 16 | 87.3 (-0.2) | 85.6 (+0.4) | 84.3 (+1.6) | 89.4 (-0.1) | 83.8 (+1.5) | 89.4 (+0.0) | 81.7 (+0.8) | 89.5 (+0.1) | 88.8 (-0.3) | 87.3 | 85.1 |
| | | | | | ImageNet-100 | | | | | | |
| 1 | 80.8 (+26.0) | 80.8 (+26.0) | 80.8 (+26.0) | 80.8 (+26.0) | 80.8 (+26.0) | 80.8 (+26.0) | 80.8 (+26.0) | 80.8 (+26.0) | 80.8 (+26.0) | 76.7 | 76.6 |
| 2 | 85.4 (+8.9) | 83.8 (+23.8) | 82.5 (+24.7) | 86.2 (+7.2) | 86.6 (+7.9) | 88.0 (+4.1) | 84.8 (+8.3) | 87.0 (+5.1) | 87.3 (+2.4) | 89.5 | 89.1 |
| 4 | 88.8 (+2.0) | 86.2 (+11.8) | 85.6 (+17.0) | 88.9 (-0.1) | 87.9 (-0.3) | 90.6 (-0.1) | 86.6 (+0.9) | 89.6 (-0.2) | 90.2 (-0.5) | 92.3 | 91.7 |
| 8 | 91.6 (+0.6) | 88.4 (+2.7) | 87.8 (+6.0) | 91.5 (-0.7) | 89.2 (-0.6) | 92.3 (-0.7) | 88.6 (-0.5) | 92.3 (-0.3) | 93.2 (-0.2) | 93.3 | 92.7 |
| 16 | 93.0 (-0.3) | 91.0 (+0.5) | 90.5 (+0.8) | 93.5 (-0.8) | 91.3 (-0.1) | 94.1 (-0.3) | 90.4 (-0.7) | 93.8 (-0.4) | 94.3 (-0.1) | 93.4 | 93.3 |
| | | | | | DomainNet-Real | | | | | | |
| 1 | 68.5 (+23.6) | 68.5 (+23.6) | 68.5 (+23.6) | 68.5 (+23.6) | 68.5 (+23.6) | 68.5 (+23.6) | 68.5 (+23.6) | 68.5 (+23.6) | 68.5 (+23.6) | 64.8 | 63.9 |
| 2 | 71.9 (+10.1) | 70.6 (+16.5) | 70.1 (+19.7) | 72.0 (+10.1) | 71.7 (+11.5) | 73.1 (+7.8) | 71.1 (+12.2) | 72.6 (+7.4) | 73.6 (+2.1) | 73.0 | 73.8 |
| 4 | 76.0 (+2.9) | 72.9 (8.4) | 72.2 (+13.5) | 75.7 (+3.3) | 73.9 (4.6) | 77.1 (1.4) | 73.7 (+4.5) | 76.6 (+1.2) | 76.2 (-0.4) | 74.8 | 77.8 |
| 8 | 79.6 (+0.4) | 77.1 (+3.0) | 76.5 (+6.2) | 80.1 (+0.2) | 77.5 (+1.2) | 81.0 (+0.3) | 77.0 (+1.1) | 80.8 (+0.1) | 78.4 (-0.3) | 76.2 | 80.6 |
| 16 | 82.2 (+0.1) | 81.3 (+0.9) | 80.7 (+1.7) | 84.0 (-0.0) | 81.2 (+0.1) | 84.0 (-0.0) | 80.5 (+0.1) | 84.3 (-0.1) | 79.5 (-0.2) | 78.0 | 82.3 |

We modify these queries to query the top-$(K \cdot B)$ samples based on their uncertainty metrics ($K = 50$) and then cluster these samples by $k$-means into $B$ clusters and select points closest to the cluster centroids. We also experiment with enabling dropout at inference time to add an element of stochasticity that disrupts the classifier's estimates of uncertainty, allowing for diverse sample selection. We report the results of our experiments in Table 2. To decouple the influence of the initial pool selection, all queries, including **Typiclust** and **ProbCover**, use an identical randomly selected pool of initial samples. Our results show that by imposing simple diversity measures, uncertainty-based queries like **Uncertainty**, **Entropy**, **Margins**, and **BALD** surpass the performance of AL strategies that explicitly incorporate diversity in their queries.

### 3.3 Representative versus uncertainty sampling and the phase transition

Much prior art in AL has placed an emphasis on the distinction between sampling the most uncertain instances in $\mathcal{D}_U$ for the active learner and sampling diverse, representative instances for establishing well-defined decision boundaries. It is a well-known fact that querying diverse but relatively representative instances early on in the AL procedure allows for the classifier to learn the structure of the representation space early in order to make uncertainty estimates in later iterations more reliable (Hacohen et al., 2022). This trade-off between diversity sampling in the low-data regime before uncertainty sampling is known as the *phase transition* (Hacohen et al., 2022).

However, when leveraging large foundation models as feature extractors, we observe a phenomenon that is a striking deviation from the existing literature. As demonstrated in Table 2, when controlling for random initialization, uncertainty sampling is actually competitive to representative sampling methods like **Typiclust** as early as the 2nd AL iteration (CIFAR100, DomainNet-Real) and even beat these methods in later iterations. Using foundation models in our diverse selection of datasets, we see no evidence of a phase transition and find no support for the notion that uncertainty sampling is ineffective in low-budget AL. Further, we witness that **Typiclust** underperforms the uncertainty-based queries in later iterations (8-16, Food101 & DomainNet-Real).

Table 2: Marginal ($\Delta$) in performance for stand-alone uncertainty measures (i.e. softmax uncertainty, predictive entropy, margins sampling, or BALD) when using clustering ($\Delta_k$) or when using clustering with dropout during inference ($\Delta_{k+d}$). For fairness, we hold all queries to use a randomly selected initial pool. Power BALD, Coreset, BADGE, Alfa-mix, Typiclust, and ProbCover inherently enforce diversity in their respective queries, so we do not show $\Delta$s for these methods. The cells are color-coded according to the magnitude of $\Delta$ for better visualization.

| $t$ | Uncertainty | | | Entropy | | | Margins | | | BALD | | pBALD | Coreset | BADGE | Alfa-mix | Typiclust | ProbCover |
| --- | Acc. | $\Delta_k$ | $\Delta_{k+d}$ | Acc. | $\Delta_k$ | $\Delta_{k+d}$ | Acc. | $\Delta_k$ | $\Delta_{k+d}$ | Acc. | $\Delta_k$ | | | | | | |
| | | | | | | | CIFAR 100 | | | | | | | | | | |
| 2 | 79.6 | +13.1 | +17.0 | 79.4 | +16.6 | +21.4 | 80.4 | +7.9 | +10.5 | 76.9 | +9.5 | 71.2 | 64.5 | 72.5 | 78.0 | 78.9 | 74.6 |
| 4 | 86.7 | +12.0 | +12.0 | 86.7 | +15.9 | +16.3 | 87.1 | +4.1 | +4.5 | 86.6 | +6.2 | 83.9 | 78.2 | 84.1 | 83.9 | 86.4 | 82.2 |
| 8 | 89.9 | +5.6 | +5.7 | 89.8 | +8.2 | +8.4 | 89.8 | +1.8 | +1.9 | 89.3 | +4.1 | 88.4 | 88.3 | 90.9 | 90.8 | 90.0 | 89.1 |
| 16 | 91.2 | +2.2 | +1.9 | 91.4 | +3.5 | +3.6 | 91.4 | +0.9 | +0.7 | 91.0 | +2.7 | 90.8 | 88.3 | 90.9 | 90.8 | 90.0 | 89.1 |
| | | | | | | | Food101 | | | | | | | | | | |
| 2 | 70.3 | +16.1 | +19.4 | 68.8 | +14.3 | +18.9 | 72.3 | +7.4 | +9.6 | 64.0 | +13.2 | 67.1 | 51.5 | 64.8 | 74.1 | 77.0 | 73.8 |
| 4 | 81.8 | +17.0 | +18.3 | 80.0 | +17.6 | +20.2 | 81.7 | +5.7 | +6.2 | 77.5 | +15.3 | 78.2 | 63.0 | 77.5 | 79.6 | 82.5 | 78.6 |
| 8 | 86.4 | +9.3 | +9.1 | 85.7 | +11.1 | +12.7 | 86.1 | +1.5 | +1.5 | 84.3 | +10.0 | 85.6 | 73.4 | 85.3 | 85.6 | 85.8 | 82.3 |
| 16 | 88.9 | +4.5 | +3.6 | 88.7 | +5.9 | +6.1 | 89.0 | +0.1 | -0.4 | 87.9 | +5.6 | 89.4 | 80.8 | 89.4 | 89.2 | 87.4 | 85.3 |
| | | | | | | | ImageNet-100 | | | | | | | | | | |
| 2 | 88.5 | +21.6 | +28.5 | 88.4 | +21.1 | +30.5 | 88.5 | +8.5 | +9.5 | 86.7 | +8.0 | 83.9 | 76.5 | 81.8 | 84.9 | 89.1 | 87.0 |
| 4 | 90.6 | +15.6 | +16.2 | 90.4 | +20.9 | +21.8 | 91.2 | +2.3 | +2.2 | 90.2 | +2.0 | 90.6 | 85.7 | 89.8 | 90.7 | 92.4 | 91.8 |
| 8 | 93.3 | +7.6 | +7.6 | 92.0 | +11.3 | +10.3 | 93.3 | +1.3 | +1.1 | 91.4 | +1.6 | 92.9 | 89.0 | 92.6 | 93.4 | 93.3 | 92.7 |
| 16 | 94.1 | +4.1 | +3.6 | 94.2 | +4.7 | +4.5 | 94.4 | +0.3 | +0.2 | 93.3 | +1.9 | 94.4 | 91.1 | 94.1 | 94.4 | 93.8 | 93.8 |
| | | | | | | | DomainNet-Real | | | | | | | | | | |
| 2 | 71.5 | +12.6 | +17.4 | 71.2 | +15.4 | +20.8 | 72.3 | +7.1 | +10.4 | 68.1 | +7.8 | 65.3 | 58.8 | 65.2 | 71.4 | 72.4 | 71.3 |
| 4 | 79.4 | +13.9 | +14.9 | 78.8 | +19.5 | +20.1 | 79.1 | +6.5 | +6.6 | 78.0 | +8.7 | 75.6 | 69.2 | 75.4 | 76.6 | 75.2 | 77.3 |
| 8 | 82.7 | +8.3 | +8.5 | 82.5 | +12.2 | +12.2 | 82.4 | +2.6 | +2.5 | 81.7 | +5.4 | 80.7 | 76.0 | 80.7 | 78.7 | 77.0 | 80.4 |
| 16 | 84.6 | +4.4 | +4.2 | 84.5 | +5.8 | +5.5 | 84.4 | +0.7 | +0.4 | 84.5 | +3.3 | 84.0 | 80.4 | 84.3 | 79.7 | 78.3 | 82.1 |

## 3.4 Leveraging unlabeled instances for training the active learner

Even with a good initial query, the classifier $g(z_i; \theta)$ may perform poorly in the low-data regime since there are a limited number of training examples. In cases where the labeling budget is prohibitive, acquiring enough training instances for a classifier to establish good decision boundaries early on in AL may be difficult. This motivates the use of unlabeled instances in a principled way to ensure that the classifier maximizes performance even in the low-data regime. However, our experiments find that the efficacy of semi-supervised learning in this setting is questionable at best, with wide variation across query methods and datasets. While we do see an initial boost in performance, contrary to Gao et al. (2020) and Simeoni et al. (2021), the gap quickly narrows, and label propagation underperforms the reference query. Propagating labels from uncertain queried samples may cause points across the decision boundary to be assigned to incorrect classes, hampering performance in the long run. We point the reader to the Appendix for an in-depth analysis using a popular semi-supervised algorithm.

## 3.5 Summary of results

Our experiments with initial pool selection (we study the importance of intelligently selecting the initial labeled pool by comparing centroid-based initialization with random initialization) motivate the use of a centroid-based initialization strategy to overcome the cold start problem. Additionally, our experiments modifying existing uncertainty-based approaches to enforce diversity in query selection, which motivates the use of clustering to select diverse samples during active learning. Our findings also indicate that a clustering approach similar to **Alfa-Mix** selects diverse candidates that span the representation space. Furthermore, our investigation into the previously identified phase transition implicates the need for an uncertainty-based query function instead of representative sampling throughout active learning. We revisited the notion that uncertainty-based queries are outperformed by representative sample selection methods such as **Typiclust** in the low-budget regime. We find that this is not the case, contradicting this existing notion. Finally, our inquiry into enhancing the active learner with unlabeled instances cautions against the use of semi-supervised learning as a complementary approach to active learning.

Based on our findings, an effective AL strategy would initialize the labeled set with representative samples, employ an uncertainty-based query function for shortlisting unlabeled candidates, and select a diverse subset of these candidates.

## 4 DropQuery , a simple, effective active learning strategy

The observations from the previous sections directly inform the principled construction of a new AL strategy leveraging the robust visual features of foundation models, which we refer to as **DropQuery** . A detailed description of the query strategy is provided in Algorithm 1. Below, we briefly review the key results from section 3 that motivate the choice of components and the construction of **DropQuery** .

- **Centroid-based initial pool selection:** Our experimental results in Section 3.1 demonstrated the utility of intelligently selecting the initial pool of candidates to label. Informed by these results and given the semantic clusters characteristic to the latent spaces of foundation models, **DropQuery** employs a centroid-based initial pool selection strategy to overcome the cold-start problem.

- **Uncertainy-based sampling approach:** Based on our investigation into the tradeoff between uncertainty-based and representative query sampling in Section 3.3, **DropQuery** favors an uncertainty-based query strategy. We experiment using dropout perturbations to measure uncertainty and select candidates for annotation. Given features $z_i$ of an unlabeled instance, we produce $M$ dropout samples of these inputs ($\rho = 0.75$) to get $\{z_i^1, \ldots, z_i^M\}$. The current classifier at $t$ then makes predictions on these $M$ samples and the original instance $z_i$, and we measure the consistency of classifier predictions $\hat{y}_i = g(z_i; \theta)$. If more than half of the $M$ predictions are inconsistent, we add $z_i$ to the candidate set $Z_c$. In all of our experiments, we set $M = 3$.

- **Selecting a diverse pool of candidates:** Our analysis in Section 3.2 provided evidence in favor of choosing a diverse set of candidates to annotate. **DropQuery** leverages this result by taking the candidate set $Z_c$ and clustering it into $B$ clusters, selecting the point closest to the center of each cluster. These constitute a diverse subset of points to be annotated.

- **Leveraging unlabeled instances:** The results from our investigation in Section 3.4 did not provide conclusive evidence in favor of adding a semi-supervised learning component to **DropQuery** , hence we choose to omit this particular variant of label propagation from our strategy.

---

**Algorithm 1** DropQuery

---

**Input:** Given unlabeled instances $z_i \in Z_U$, external oracle $\phi(.)$, budget $B$.
**Output:** Queried labels $Y = \{y_i : i \in 1, \ldots, B\}$
  **for** $z_i \in Z$ **do**
    $\{z_i^1, \ldots, z_i^M\} \leftarrow \text{Dropout}(z_i; \rho)$                 ▷ Apply dropout to the input features
    $n_i \leftarrow \sum_{m=1}^{M} \mathbb{1}[g(z_i^m; \theta) = g(z_i; \theta)]$      ▷ Measure inconsistency among classifier predictions
  **end for**
  $Z_c = \{z_i | n_i > 0.5M \; \forall z_i \in Z_U\}$            ▷ Use consistency as proxy for uncertainty
  $C_k \leftarrow K\text{-means}(Z_c, B)$ where $k \in \{B\}$         ▷ Cluster points to enforce diversity
  $S_k \leftarrow \{z_i : \arg\min_{z_i \in Z_c} \|z_i - c_k\|_2^2 \text{ where } c_k \in C_k\}$
  $Y \leftarrow \phi(S_k)$                   ▷ Retrieve labels from samples closest to centroids

---

We take inspiration from works like Ducoffe & Precioso (2017) and Beluch et al. (2018), which leverage query-by-committee to distinguish informative samples. It is well-known that dropout on the weights of a classifier during inference simulates an ensemble of models (Warde-Farley et al., 2014). However, our strategy is a nuanced but critical change from these works. **DropQuery** is more similar in spirit to ALFA-Mix, which interpolates unlabeled points with anchors to perturb them and tests the consistency of their class predictions. Applying dropout perturbs features, and those lying close to the decision boundary will

have inconsistent predictions, making them good candidates for querying. Candidate clustering ensures the diversity of selected points, as shown in Table 3. Since label propagation does not help queries in our experiments, we do not consider it a core component.

Table 3: Test accuracy for our method (with random acquisition as a reference) utilizing dropout at inference time (DQ), with a centroid-based initial pool ($DQ_c$), and dropout with semi-supervised learning in the form of label propagation ($DQ_{ssl}$).

| $t$ | Random | DQ | $DQ_c$ | $DQ_{ssl}$ | Random | DQ | $DQ_c$ | $DQ_{ssl}$ | Random | DQ | $DQ_c$ | $DQ_{ssl}$ | Random | DQ | $DQ_c$ | $DQ_{ssl}$ |
|---|---|---|---|---|---|---|---|---|---|---|---|---|---|---|---|---|
| | CIFAR100 | | | | ImageNet-100 | | | | Food101 | | | | DomainNet-Real | | | |
| 1 | 48.1 | 48.1 | 72.6 | 53.3 | 54.7 | 54.7 | 79.1 | 57.8 | 47.4 | 47.4 | 71.3 | 49.7 | 44.8 | 44.8 | 68.0 | 46.8 |
| 2 | 64.5 | 81.0 | 83.5 | 79.2 | 76.5 | 88.8 | 89.4 | 85.9 | 64.4 | 72.7 | 76.1 | 70.6 | 61.8 | 73.0 | 75.0 | 68.9 |
| 4 | 78.7 | 87.4 | 87.7 | 87.1 | 86.8 | 91.7 | 91.5 | 90.9 | 77.2 | 81.8 | 82.2 | 80.9 | 73.1 | 79.1 | 78.7 | 76.7 |
| 8 | 86.1 | 89.8 | 90.1 | 89.6 | 91.0 | 93.2 | 93.0 | 92.8 | 83.8 | 86.3 | 86.3 | 84.7 | 79.2 | 82.4 | 82.2 | 80.0 |
| 16 | 89.2 | 91.4 | 91.5 | 91.0 | 93.3 | 94.1 | 94.1 | 93.7 | 87.5 | 89.5 | 89.4 | 86.7 | 82.1 | 84.8 | 84.7 | 82.1 |

# 5    Experimental results

Our strategy's effectiveness is demonstrated on several natural, out-of-domain biomedical, and large-scale image datasets. Initially, we assess **DropQuery** 's performance, maintaining the dataset constant while altering the underlying representation space (see Figure 1). Our experiments confirm that larger models generate more resilient visual features, thereby enhancing linear classifier performance in low-budget scenarios. An intriguing phenomenon, observed during iterations 4-16 in Figure 1, reveals that *smaller* models produce embeddings that our AL strategy better accommodates in later iterations. The performance benefits of larger models in early iterations (limited data availability) diminish rapidly, likely due to their expressiveness compared to smaller models' features. Hence, dropout-motivated uncertainty sampling is not as advantageous. We further note that, on average, vision-language foundation models outperform their vision-only counterparts.

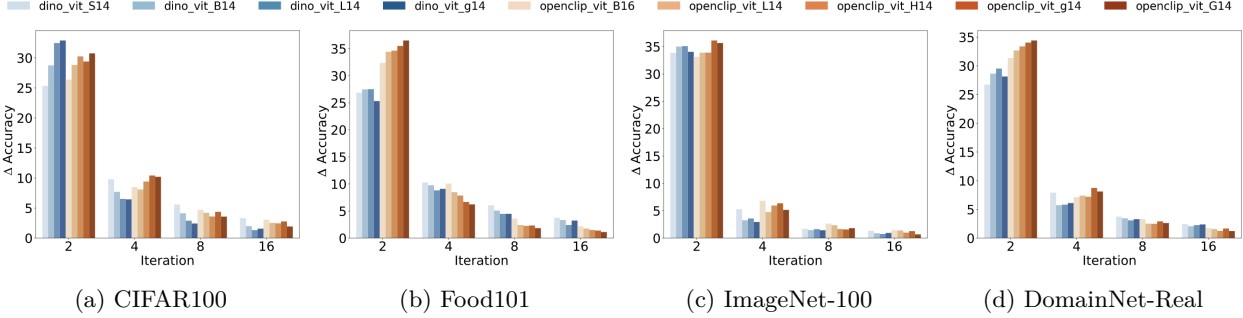

Figure 1: Results of our AL strategy on different representation spaces (i.e. DINOv2 (Oquab et al., 2023) and OpenCLIP (Cherti et al., 2022)). The y-axis is the delta in accuracy between iteration $i$ and $i/2$. In early iterations, the improvements to AL query performance are more pronounced for larger models.

## 5.1    Natural Image Datasets

Our proposed AL strategy is evaluated through a comprehensive set of experiments on diverse natural image datasets sourced from the VTAB+ benchmark (Schuhmann et al., 2022). The VTAB+ benchmark, an expanded version of the VTAB (Zhai et al., 2020) benchmark, encompasses a range of challenging visual perception tasks including fine-grained image classification and is typically used for assessing zero-shot performance. A notable drawback in many prior AL studies is the excessive reliance on datasets such as CIFAR100 or TinyImageNet with small image sizes that are not representative of real-world scenarios where AL would be necessary. Therefore, our experiments include several larger, more complex natural image

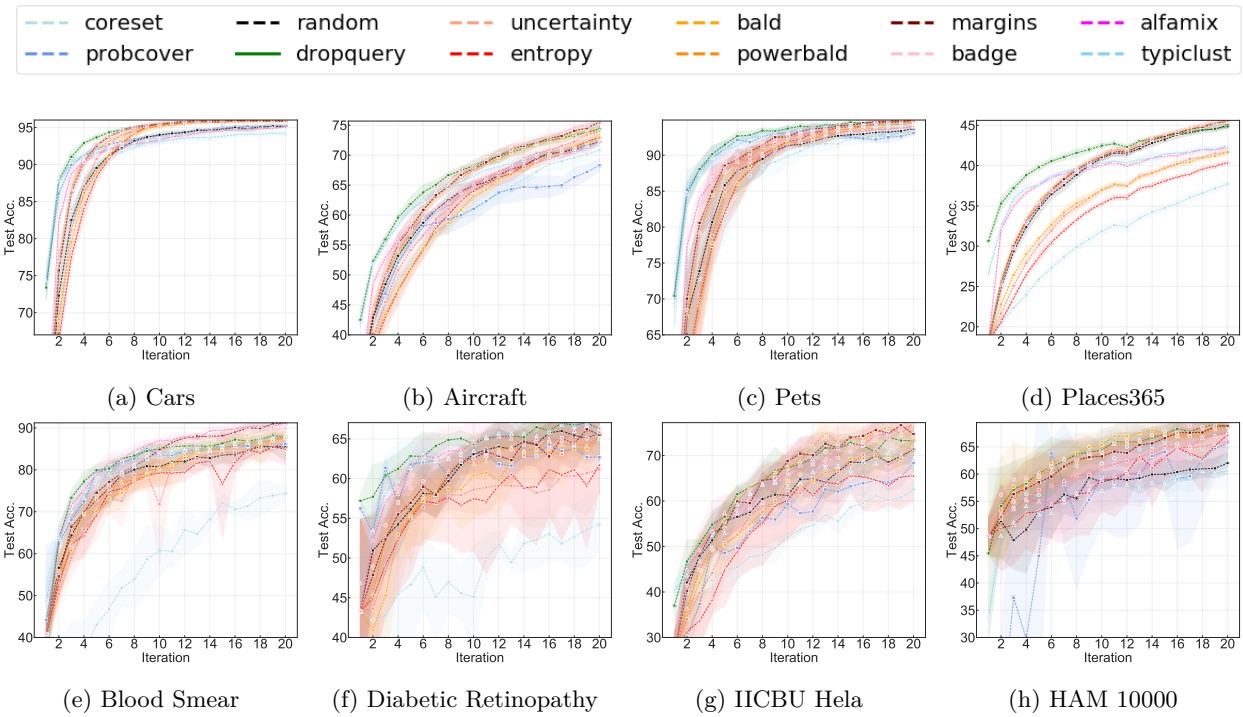

Figure 2: (Top row) Results on fine-grained natural image classification tasks Stanford Cars (Krause et al., 2013), FVGC Aircraft (Maji et al., 2013), Oxford-IIIT Pets (Parkhi et al., 2012), and the Places365 (**?**) datasets. (Bottom row) AL curves for biomedical datasets, including images of peripheral blood smears (Acevedo et al., 2020), retinal fundoscopy (Kaggle & EyePacs, 2015), HeLa cell structures (Murphy et al., 2000), and skin dermoscopy skin (Tschandl et al., 2018), covering pathology, ophthalmology, cell biology, and dermatology domains using various imaging modalities. Additional biomedical datasets are explored in the Appendix.

datasets that closely mirror real-world scenarios. We describe extensive implementation and training details in the Appendix.

Among the fine-grained natural image classification datasets, in Stanford Cars (Krause et al., 2013) and Oxford-IIIT Pets (Parkhi et al., 2012), our **DropQuery** outperforms all other AL queries in each iteration while also outperforming complex query approaches like **Alfa-Mix** and **Typiclust** in the low-budget regime in FVGC Aircraft (Maji et al., 2013) (see Figure 2). Our approach, which is agnostic to dataset and model, outperforms the state-of-the-art AL queries, which often necessitate additional hyperparameter tuning given the underlying data distribution. We also test our method on a large-scale dataset with 365 classes, Places365 (Zhou et al., 2017) (which contains approximately 1.8 million images), and our strategy beats all other modern AL queries. These results exemplify the scalability of our method on large datasets where AL would be used to probe the unlabeled data space for important samples.

## 5.2 Out-of-domain datasets

In addition, we conduct experiments on various biomedical datasets, which significantly deviate from the training distribution of the foundation models discussed in this study. The aim of these experiments is to underscore the effectiveness of AL queries when applied to datasets that are out-of-domain or underrepresented in pretraining. This kind of evaluation is especially pertinent in situations where a model trained in one institution or setting is deployed in another. An efficient AL query is helpful to obtain a minimum number of samples to label to fine-tune the model for the new task — a realistic scenario often neglected in previous studies. 2 illustrates our strategy's performance on challenging out-of-domain data. In real-world contexts,

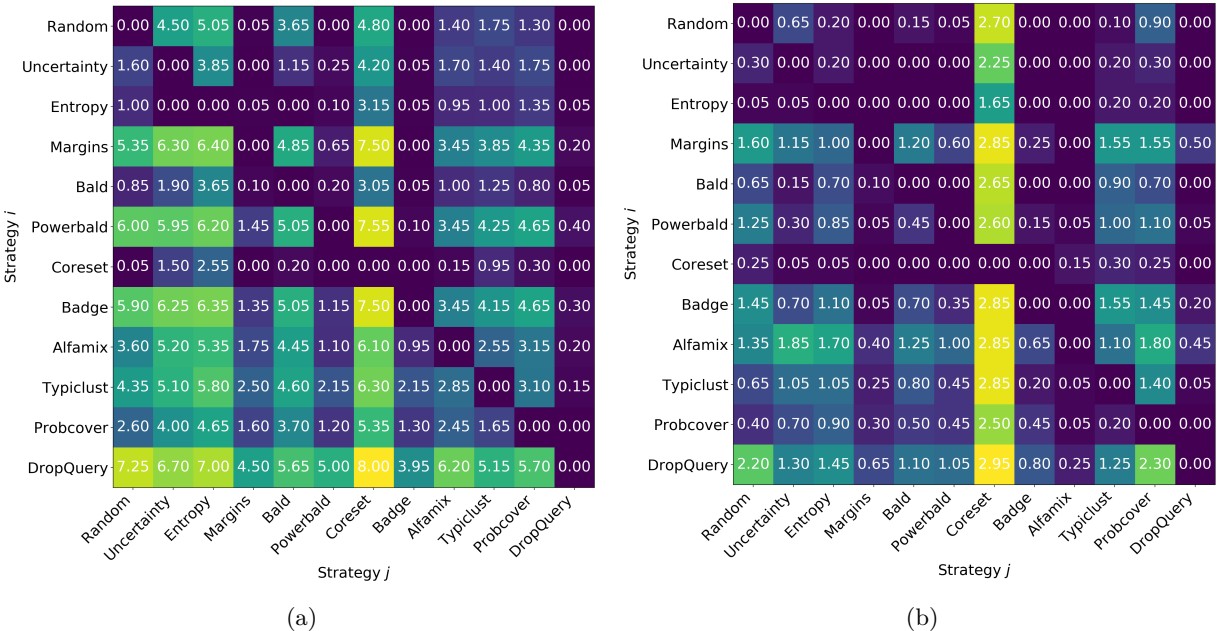

Figure 3: Win matrices for all AL strategies investigated in our study evaluated on natural image datasets and out-of-domain biomedical image datasets. **(a)** CIFAR100, Food101, Imagenet-100, and DomainNet-Real using DINOv2 VIT-g/14 features and Stanford Cars, FVGC Aircraft, Oxford-III Pets, and Places365 using OpenCLIP VIT-G/14 features (8 total settings). Due to computational costs, **ProbCover** was not evaluated on Places365, so the max value of cells in the **ProbCover** row/column is 7. **(b)** Blood Smear, Diabetic Retinopathy, IICBU Hela, and Skin cancer datasets using DINOv2 VIT-g/14 features (4 total settings). **DropQuery** outperforms all other methods on the natural image datasets and is a strong competitor to all other methods on the biomedical image datasets with statistical significance.

**DropQuery** excels as a practical, efficient method for querying hard-to-classify samples for improving the model.

### 5.3 Statistical significance study

In order to verify the empirical strength of **DropQuery** , we furnish our results with statistical significance tests following the procedure in Parvaneh et al. (2022) and Ash et al. (2020). We conduct paired $t$-tests to determine whether a particular AL strategy $i$ outperforms another strategy $j$. We say that query strategy $i$ beats or surpasses strategy $j$ at iteration $r$ if

$$c_{i,j}^r = \frac{\sqrt{5}\mu^r}{\sigma^r} > 2.776, \quad \mu^r = \frac{1}{5}\sum_{k=1}^{5}(a_i^r - a_j^r), \quad \sigma^r = \sqrt{\frac{1}{5}\sum_{k=1}^{5}(a_i^r - a_j^r - \mu^r)^2}$$

where $a_i^r$ is the accuracy of strategy $i$ at iteration $r$. This corresponds to the $p = 0.05$ threshold for significance. We compute these $c_{i,j}^r$ for all iterations and sum the number of times strategy $i$ surpassed strategy $j$ at each iteration $r$ (divided by the total number of iterations) to indicate the score of $i$ beating $j$ in a specific dataset-model setting. Figure 3 illustrates two "win" matrices illustrating the proportion of times strategy i has a significantly higher (p<0.05) mean accuracy compared to strategy j. For each pair, at each iteration, the mean accuracy for methods i and j are compared using a t-test with p<0.05, and the outcome is win, lose, or tie to yield a fraction of wins from (0, 1). This is repeated for all datasets considered and the values are summed to yield the proportion of wins. For example, in Figure 3a, **DropQuery** beats random sampling in 7.25/8 fraction of total AL iterations across the 8 dataset settings.

### 5.4 Ablations for DropQuery

### 5.4.1 Impact of the dropout ratio

We explore the influence of the dropout ratio on both the overall performance of the models and the number of selected candidates at each iteration step. We experiment with a range of values - [**0.15**, **0.30**, **0.45**, **0.60**] in addition to **0.75**, the setting used in our main experiments. Our findings, highlighted in Appendix Table 1, indicate that higher dropout ratios for the features result in slightly better-performing models, a consistent trend among datasets. This is perhaps unsurprising, as stronger regularization in the form of harsher dropout ratios would improve the generalizability of models in the low-budget regime. Interestingly, lower dropout ratios lead to fewer samples added to the candidate set. This may be a complementary factor affecting the performance of the model, as a smaller candidate set would likely be less diverse.

### 5.4.2 Impact of $M$, the number of dropout iterations

The number of dropout iterations, $M$, is a tunable hyper-parameter for our query function. We experiment with a range of values - [**5**, **7**, **9**] in addition to **3**, the value used in our main experiments, and present the results in Appendix Table 2. Our findings indicate that this hyper-parameter has a minimal impact on both the performance of the model and the number of selected candidates, with fewer iterations having a slight advantage. The robustness of our models' performance w.r.t $M$ is a desirable trait for an AL query function designed for the low-budget regime since the existence of a validation set for hyper-parameter tuning cannot be assumed due to the paucity of labels.

Over the course of our experiments, we found that other query functions could perform better with the right combination of hyperparameters. ProbCover performed better with lower purity levels, and ALFA-Mix performed better with a lower value of $\epsilon$. The maximum number of clusters was set arbitrarily for TypiClust, as was the number of neighbors used to calculate typicality. Ideally, these hyper-parameters would have been determined based on the distribution of features, the number of dimensions, and the number of samples in each dataset. These observations strengthen our belief that DropQuery is a strong competitor to existing methods due to its simplicity and excellent performance without tuning $M$.

## 6 Conclusions

In this work, we systematically study four critical components of effective active learning in the context of foundation models. Given our observations, we propose a new AL strategy, named DropQuery , that combines the benefits of strong initial pool selection by class centroids and a dropout-motivated uncertainty measure for querying unlabeled instances. This method outperforms other AL queries in a variety of natural image, biomedical, and large-scale datasets. It poses a paradigm shift to leveraging the properties of foundation model representations in AL.

**Limitations and Societal Impact** Our work is a principled investigation focused on the interplay of AL and foundation models, and we recognize certain limitations. Firstly, our approach presumes the availability of public foundation models, which may not be universally accessible. Secondly, inherent biases and ethical dilemmas tied to these models could be reflected in our method, possibly leading to biased outcomes or the propagation of harmful stereotypes. Thirdly, our experiments focus on datasets with a relatively even distribution of labels, as has been the norm with most works exploring AL queries. Given that long-tailed datasets are quite common in the real world, there is a concern that our findings may not generalize to scenarios with heavily imbalanced class labels. This limitation warrants further investigation, particularly in the biomedical domain, where class imbalances and long-tailed distributions are widespread. Despite these issues, the growth of foundation models encourages their use in AL, even with out-of-domain datasets. Finally, we acknowledge that the semi-supervised learning method implemented in this study is not the only way to leverage unlabeled instances in this manner. Due to our experimental setup involving a fixed representation space generated by frozen foundation model backbones, we are constrained to a subset of SSL algorithms that operate in an offline mode. However, we encourage future work to investigate further along these lines and emphasize that our recommendation is limited to the use of a particular label propagation algorithm. We urge future research to address these concerns while considering the broader ethical implications.

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
