# Supplementary for: Revisiting Active Learning in the Era of Vision Foundation Models

**Sanket Rajan Gupte**                                           *sanketg@stanford.edu*
*Department of Computer Science*
*Stanford University*

**Josiah Aklilu**                                                *josaklil@stanford.edu*
*Department of Biomedical Data Science*
*Stanford University*

**Jeffrey J. Nirschl**                                           *jnirschl@stanford.edu*
*Department of Pathology*
*Stanford University*

**Serena Yeung-Levy**                                            *syyeung@stanford.edu*
*Department of Biomedical Data Science*
*Stanford University*

**Reviewed on OpenReview:** *https://openreview.net/forum?id=u8K83M9mbG*

## A  Appendix

### A.1  Implementation details

Our AL framework is implemented using PyTorch Lightning Falcon & The PyTorch Lightning team (2019) for training and evaluation of the models and uses Faiss Johnson et al. (2019) for GPU-accelerated clustering and nearest-neighbor searches. DINOv2 pre-trained models have been downloaded from https://github.com/facebookresearch/dinov2 and OpenCLIP pre-trained models are loaded using https://github.com/mlfoundations/open_clip. Our implementation is available at https://github.com/sanketx/AL-foundation-models.

We conduct AL experiments for 20 iterations with 5 seeds - 1,10,100,1000,10000 for each combination of query, dataset, and model that we evaluate and report the mean accuracy averaged over all 5 seeds. The query budget at each iteration is set to $C$ where C is the number of classes in the dataset. All linear classifiers in our experiments are trained using the AdamW optimizer with a learning rate of 1e-2, weight decay of 1e-2, and dropout with $\rho = 0.75$. In experiments involving label propagation, we set $\alpha = 0.9$ and construct the graph using the 500 nearest neighbors.

Computations are carried out on A100 (40GB) GPUs, however we note that GPUs with 16-24GB of memory will also suffice for the majority of our experiments, with each individual run of 20 iterations taking approximately 15-30 minutes.

### A.1.1  AL query implementations

All the AL queries tested in our experiments have been reimplemented to optimize them for compute and memory efficiency. BALD and PowerBALD both use 20 MC sampling iterations for uncertainty estimation. Core-Set is implemented using the greedy k-center approach. ALFA-Mix is implemented using the closed-form approximation of $\alpha$ with $\epsilon = \frac{0.2}{\sqrt{D}}$ where $D$ is the dimensionality of $\alpha$. For TypiClust, we set the maximum number of clusters to 500 for all datasets except Places365 where we set it to 1000. Typicality is calculated using the 20 nearest neighbors. $\delta$ in ProbCover is estimated with a purity threshold of 0.95.

BADGE is difficult to scale to large datasets such as Places365 due to the size of the gradient embedding vectors. They have $N \times C \times D$ elements where $N$ is the number of unlabeled samples, $C$ is the number of classes, and $D$ is the dimensionality of the output of the penultimate layer, which in our case is feature vector derived from the foundation model. Fortunately, the K-Means++ initialization scheme used to pick diverse points only requires the computation of the squared distance between pairs of gradient embeddings, so the full embeddings themselves need not be computed.

The squared distance between a pair of gradient embeddings $G_i, G_j$ can be expressed in terms of their squared Frobenius norm $||G_i - G_j||_F^2$ where $G_i = Z_i P_i^T$. Here, $Z_i$ is the $D$ dimensional feature vector input to the linear classifier and $P_i = (p_k^{(i)} - I(\hat{y}^{(i)} = k))_{k=1}^C$, the difference of the predicted probability vector and the one hot encoded pseudo-label of the $i$th sample.

$||G_i - G_j||_F^2$ can be efficiently factorized as $Z_i^T(P_i^T P_i)Z_i + Z_j^T(P_j^T P_j)Z_j - 2Z_i^T(P_i^T P_j)Z_j$ which only requires $O(N \cdot (C + D))$ space instead of $O(N \cdot C \cdot D)$, enabling BADGE to scale to datasets with several million unlabeled examples.

## A.2 Utilizing unlabeled instances for improving the active learner

In this section, we explore leveraging unlabeled instances in a semi-supervised fashion as a complementary approach to active learning. Prior art such as variants of **Typiclust** and **ProbCover** use Flex-Match Zhang et al. (2022), a state-of-the-art semi-supervised learning algorithm based on consistency among augmented views of an image. However, this is challenging to implement in our context as repeated forward passes through the feature extractor would be computationally expensive. Since foundation models are typically trained in a self supervised fashion using a contrastive loss (or a variant) to encourage consistency, augmented views would map to similar points in the feature space.

Several works have suggested using state-of-the-art semi-supervised frameworks for label propagation from labeled instances Gao et al. (2020); Simeoni et al. (2021). We adopt an offline variant of a transductive label propagation method Iscen et al. (2019). Our experimental setup accounted for the confidence of the propagated labels while training the classifier. We used equation 11 from Iscen et al. (2019) to weight each sample by using entropy as a measure of confidence. The weight was computed as $w_i = 1 - \frac{H(\hat{z}_i)}{log(c)}$. Since our representation space does not change, we only perform a cycle of label propagation when new labels are added to the pool, i.e. after each AL query. In theory, a semantically meaningful representation space from a Foundation Model would enable labels to propagate more effectively, thus increasing the efficacy of the active learner. However, our experimental results say otherwise.

We show the effects of label propagation in Figure 1 while using a randomly initialized labeled pool for all queries. We find that the efficacy of semi-supervised learning in this setting is questionable at best, with wide variation across query methods and datasets. While we do see an initial boost in performance, contrary to Gao et al. (2020); Simeoni et al. (2021), the gap quickly narrows and label propagation under performs the reference query. Propagating labels from uncertain queried samples may cause points across the decision boundary to be assigned to incorrect classes, hampering performance in the long run.

## A.3 Full results on natural image and out-of-domain data

We also tabulate full AL empirical results in Table 4, Table 5, and Table 6 for the datasets presented in the main manuscript (12 in total) and additional out-of-domain datasets like colorectal histology Kather et al. (2016) and Patch Camelyon Veeling et al. (2018); Ehteshami Bejnordi et al. (2017) (Figure 2). We repeated acquisition trajectories with 5 different random seeds for precise evaluation and report mean accuracy and standard deviation on the held out test set. **DropQuery** performs competitively on the colorectal histology dataset, but underperforms others on the Patch Camelyon dataset. The abnormal spread in performances of AL strategies on Patch Camelyon suggests that Foundation Model features may not equally benefit all approaches on out-of-distribution data, and warrants further investigation of low-budget AL in biomedical imaging.

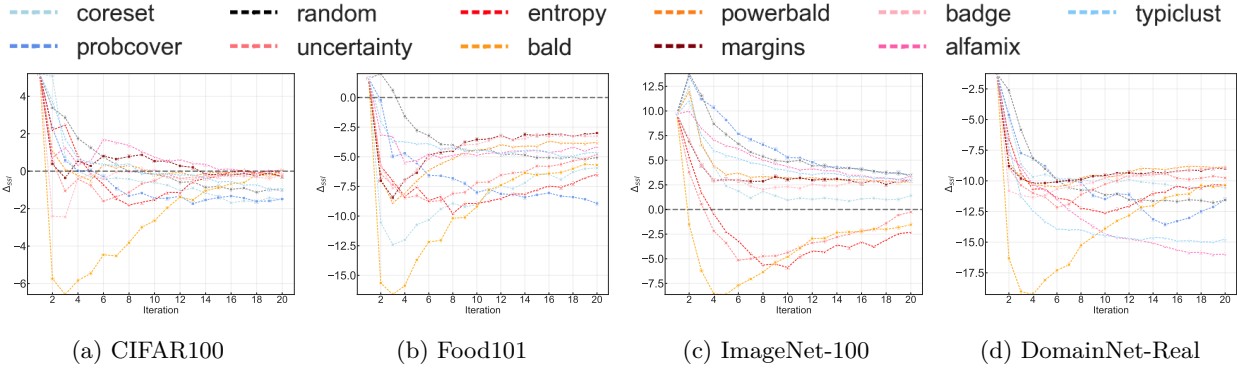

| (a) CIFAR100 | (b) Food101 | (c) ImageNet-100 | (d) DomainNet-Real |

Figure 1: We illustrate the performance difference $\Delta_{ssl}$ between AL with and without label propagation for unlabeled instances. The results, averaged over 5 runs of 20 AL iterations on 4 natural image datasets, show that the suitability of foundation models for pseudo-label approaches is, although significant in the initial iterations of AL, hurts the performance of the active learner in later iterations.

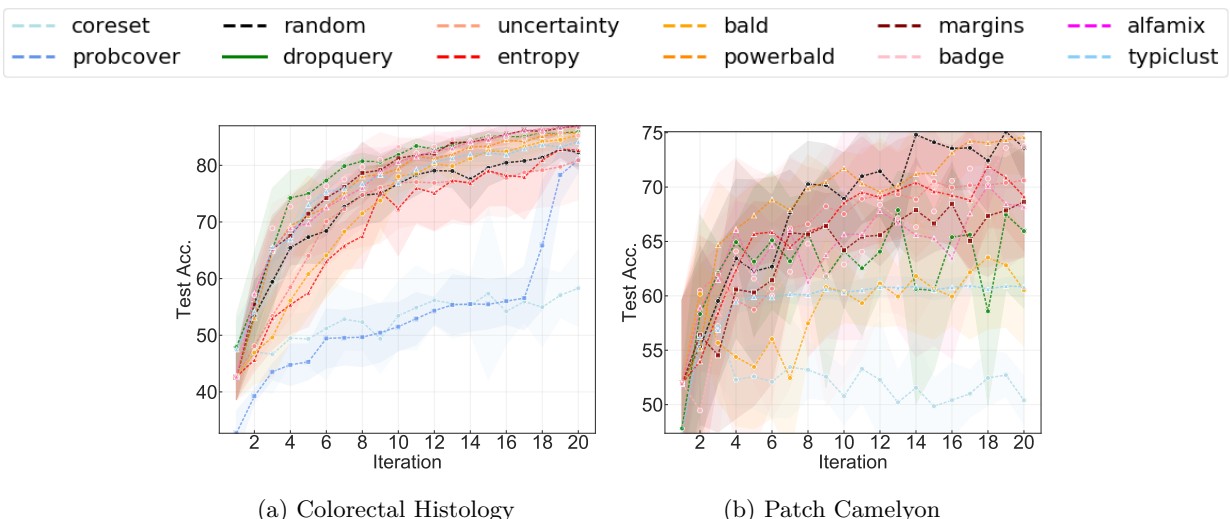

| (a) Colorectal Histology | (b) Patch Camelyon |

Figure 2: Full AL curves for additional out-of-domain biomedical datasets.

Biomedical datasets are ideal for testing active learning methods, given the time constraints and high costs of labeling by domain experts. Acevedo et al 2020's blood smear dataset Acevedo et al. (2020) consists of bright-field microscopy images of eight peripheral blood cell classes (imbalanced classes, minority/majority ratio = 0.36). The colorectal histology dataset Kather et al. (2016) includes H&E stained brightfield microscopy images of 8 different classes of textures seen in colorectal cancer histology (balanced, min/Maj ratio = 1.0). The Kaggle Diabetic Retinopathy challenge dataset Kaggle & EyePacs (2015) presents retinal images categorized into five diabetic retinopathy severity levels (imbalanced, min/Maj ratio = 0.19). The Robert Murphy lab's IICBU 2008 HeLa dataset Murphy et al. (2000) includes fluorescence microscopy images of HeLa cells, with ten different classes of labeled subcellular structures (min/Maj ratio = 0.74). The HAM10000 dataset Tschandl et al. (2018) contains dermatoscopic images across seven skin lesion classes (imbalanced, min/Maj ratio = 0.01). Lastly, the patch camelyon dataset (Veeling et al., 2018) (Ehteshami Bejnordi et al., 2017) consists of 327,680 image patches from lymph notes with the goal of binary classification of the presence or absence of metastatic breast carcinoma cells (balanced, min/Maj ratio = 1.0). The patch camelyon train/val/test splits are 262,144/ 32,768/ 32,768 respectively. Together, these datasets span fields such as cell biology, cytology, dermatology, and ophthalmology, offering a robust out-of-distribution test for active learning strategies that utilize foundation models pre-trained on natural images.

## A.4 Additional pretraining strategies

We study the impact of limited pretraining and a relatively weak feature extractor on the performance of leading AL strategies. In this experiment, we explore the use of Masked Autoencoders (MAE) He et al. (2022) pre-trained on ImageNet-1K Deng et al. (2009) as the backbone model, specifically the largest variant: ViT-H/14. ImageNet is a considerably smaller dataset compared to LVD-142M used to train DINOv2 or LAION-2B used to train OpenCLIP models and is limited in terms of the diversity of images. Representations learned with limited pretraining data may not result in well structured latent spaces like those induced by foundation models trained on hundreds of millions of images, adversely impacting the performance of AL strategies which require robust representations. Furthermore, the MAE backbone is trained using a patch-level reconstruction objective instead of a contrastive method. This encourages the model to learn strong local features, but suboptimal global features which are necessary for instance discrimination or classification tasks such as the ones studied in our experimental setting Huang et al. (2023).

We analyze the best performing AL strategies in our previous experiments, **BADGE**, **Alfa-Mix**, **Typiclust**, **Margins**, and **PowerBALD** and compare them with **DropQuery** and a random sampling baseline on the CIFAR100, Food101, ImageNet-100, and DomainNet-Real datasets. We use the same experimental conditions and hyper-parameter settings as our previous experiments and report the results in Table 7. We observe a clear trend across datasets: AL strategies which rely on clustering images in the representation space are outperformed by those which are independent of the underlying latent structure. **BADGE** performs well consistently, followed by **PowerBALD** and **Margins**, while **Typiclust**, **Alfa-Mix**, and **DropQuery** demonstrate weaker performance.

Unlike our experiments with foundation models, we do not see a significant impact of intelligent initial pool selection to overcome the cold start problem, and any advantage that **Typiclust** and **DropQuery** may have is quickly eroded. Furthermore, we observe that the MAE features do not generalize well to CIFAR100 and Food101, with even the best AL methods not performing much better than random sampling. This is likely because ImageNet-100 is a subset of the MAE pretraining dataset and is compositionally similar to DomainNet-Real. Given that the MAE backbone typically requires domain-specific fine-tuning, it is unsurprising that these features perform poorly in the extremely low-budget AL regime, especially with a simple linear classifier. We conclude by re-iterating our fundamental premise: in the era of vision foundation models, we need to revisit previous findings in active learning to advance the development of strategies that can fully leverage the rich representations generated by these powerful backbones.

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

Table 1: Impact of the dropout ratio. We evaluate 5 different dropout ratios $\rho$ for on 4 natural image datasets - CIFAR100, Food101, ImageNet-100, and DomainNet-Real. We report the mean accuracy averaged over 5 runs along with the standard deviation at each AL iteration. **Bold** values represent the **first place** mean accuracy at iteration $t$ with the second place value underlined. We also report the corresponding fraction of the unlabeled examples added to the candidate set.

| $t$ | $\rho = 0.15$ Mean ± Std | Fraction | $\rho = 0.3$ Mean ± Std | Fraction | $\rho = 0.45$ Mean ± Std | Fraction | $\rho = 0.6$ Mean ± Std | Fraction | $\rho = 0.75$ Mean ± Std | Fraction |
|---|---|---|---|---|---|---|---|---|---|---|
| | | | | | **CIFAR100** | | | | | |
| 1 | **72.93 ± 1.93** | 0.0% | 72.92 ± 1.9 | 0.0% | 72.84 ± 2.06 | 0.0% | 72.81 ± 2.1 | 0.0% | 72.58 ± 2.18 | 0.0% |
| 2 | 82.46 ± 0.59 | 4.05% | 82.15 ± 1.33 | 6.7% | 83.0 ± 1.03 | 9.9% | **83.56 ± 1.31** | 14.53% | 83.52 ± 0.76 | 21.53% |
| 3 | 85.13 ± 0.48 | 2.09% | 85.4 ± 1.18 | 3.51% | 85.49 ± 0.69 | 5.16% | 85.88 ± 0.94 | 7.84% | **86.6 ± 0.78** | 11.7% |
| 4 | 86.96 ± 0.56 | 1.76% | 86.68 ± 0.94 | 2.84% | 87.33 ± 0.53 | 4.25% | 87.15 ± 0.82 | 6.14% | **87.67 ± 0.57** | 9.22% |
| 5 | 87.81 ± 0.44 | 1.55% | 88.07 ± 0.64 | 2.59% | 88.43 ± 0.4 | 3.62% | 88.02 ± 0.75 | 5.48% | **88.94 ± 0.31** | 8.33% |
| 6 | 88.53 ± 0.39 | 1.49% | 88.67 ± 0.41 | 2.41% | 89.28 ± 0.22 | 3.39% | 89.14 ± 0.16 | 5.11% | **89.35 ± 0.16** | 7.72% |
| 7 | 88.99 ± 0.15 | 1.46% | 89.07 ± 0.41 | 2.32% | 89.59 ± 0.08 | 3.2% | 89.54 ± 0.17 | 4.84% | **89.65 ± 0.31** | 7.37% |
| 8 | 89.58 ± 0.2 | 1.42% | 89.37 ± 0.58 | 2.18% | 89.89 ± 0.22 | 3.04% | 89.88 ± 0.29 | 4.6% | **90.12 ± 0.14** | 7.07% |
| 9 | 89.76 ± 0.22 | 1.37% | 89.82 ± 0.49 | 2.21% | **90.26 ± 0.24** | 3.03% | 90.12 ± 0.17 | 4.41% | 90.23 ± 0.21 | 6.88% |
| 10 | 90.18 ± 0.28 | 1.31% | 90.14 ± 0.33 | 2.09% | 90.41 ± 0.1 | 2.92% | 90.38 ± 0.2 | 4.36% | **90.48 ± 0.22** | 6.75% |
| 11 | 90.24 ± 0.22 | 1.25% | 90.25 ± 0.27 | 2.06% | 90.62 ± 0.1 | 2.9% | **90.74 ± 0.21** | 4.16% | 90.6 ± 0.28 | 6.68% |
| 12 | 90.4 ± 0.33 | 1.24% | 90.58 ± 0.25 | 2.03% | 90.71 ± 0.22 | 2.84% | **90.86 ± 0.26** | 4.12% | 90.8 ± 0.27 | 6.54% |
| 13 | 90.58 ± 0.2 | 1.23% | 90.81 ± 0.22 | 2.03% | 90.91 ± 0.14 | 2.82% | 90.94 ± 0.2 | 4.05% | **90.99 ± 0.2** | 6.37% |
| 14 | 90.69 ± 0.27 | 1.2% | 90.97 ± 0.25 | 1.93% | 91.03 ± 0.15 | 2.77% | 91.17 ± 0.2 | 4.0% | **91.26 ± 0.25** | 6.35% |
| 15 | 90.92 ± 0.25 | 1.18% | 91.13 ± 0.26 | 1.92% | 91.2 ± 0.18 | 2.66% | 91.13 ± 0.19 | 3.99% | **91.32 ± 0.26** | 6.27% |
| 16 | 91.3 ± 0.1 | 1.19% | 91.28 ± 0.18 | 1.86% | 91.28 ± 0.15 | 2.69% | 91.42 ± 0.23 | 3.97% | **91.47 ± 0.16** | 6.0% |
| | | | | | **Food101** | | | | | |
| 1 | 70.83 ± 1.2 | 0.0% | 70.86 ± 1.17 | 0.0% | 71.29 ± 1.16 | 0.0% | 71.28 ± 1.36 | 0.0% | **71.33 ± 1.38** | 0.0% |
| 2 | 74.24 ± 2.46 | 1.45% | 75.18 ± 1.64 | 2.48% | 75.16 ± 1.72 | 3.72% | 74.8 ± 1.75 | 5.54% | **76.09 ± 1.34** | 8.49% |
| 3 | 77.6 ± 1.08 | 1.61% | 78.52 ± 1.38 | 2.73% | 78.42 ± 1.43 | 3.88% | 78.61 ± 1.48 | 5.87% | **79.76 ± 0.88** | 8.6% |
| 4 | 79.98 ± 1.2 | 1.65% | 81.35 ± 1.25 | 2.79% | 80.83 ± 1.19 | 3.99% | 81.32 ± 0.91 | 5.79% | **82.24 ± 0.52** | 8.75% |
| 5 | 82.43 ± 0.65 | 1.75% | 82.85 ± 0.58 | 2.84% | 82.8 ± 0.56 | 4.06% | 83.12 ± 1.08 | 5.87% | **83.75 ± 0.84** | 8.92% |
| 6 | 83.17 ± 0.46 | 1.69% | 84.64 ± 0.52 | 2.77% | 84.46 ± 0.25 | 4.11% | 84.35 ± 0.64 | 5.71% | **84.71 ± 0.58** | 8.58% |
| 7 | 84.53 ± 0.66 | 1.7% | 85.52 ± 0.46 | 2.75% | 85.22 ± 0.34 | 4.01% | 85.54 ± 0.78 | 5.6% | **85.72 ± 0.57** | 8.79% |
| 8 | 85.37 ± 0.78 | 1.66% | 86.12 ± 0.54 | 2.75% | 85.92 ± 0.39 | 4.08% | 86.27 ± 0.71 | 5.6% | **86.31 ± 0.4** | 8.75% |
| 9 | 85.83 ± 0.77 | 1.72% | 86.67 ± 0.38 | 2.78% | 86.49 ± 0.18 | 4.12% | **86.89 ± 0.17** | 5.83% | 86.87 ± 0.29 | 8.84% |
| 10 | 86.64 ± 0.65 | 1.66% | 87.27 ± 0.37 | 2.83% | 86.88 ± 0.15 | 4.12% | **87.38 ± 0.92** | 5.78% | 87.27 ± 0.45 | 8.96% |
| 11 | 87.13 ± 0.86 | 1.64% | 87.64 ± 0.3 | 2.77% | 87.56 ± 0.46 | 4.12% | **87.77 ± 0.73** | 5.91% | 87.72 ± 0.55 | 8.8% |
| 12 | 87.4 ± 0.75 | 1.68% | 88.02 ± 0.49 | 2.79% | 88.05 ± 0.43 | 4.1% | **88.17 ± 0.56** | 5.86% | 88.07 ± 0.63 | 8.77% |
| 13 | 87.74 ± 0.84 | 1.7% | 88.45 ± 0.5 | 2.8% | 88.15 ± 0.51 | 4.07% | 88.48 ± 0.64 | 5.93% | **88.51 ± 0.59** | 8.9% |
| 14 | 88.03 ± 0.74 | 1.71% | 88.89 ± 0.3 | 2.77% | 88.5 ± 0.34 | 4.12% | 88.82 ± 0.49 | 5.93% | **88.92 ± 0.67** | 8.94% |
| 15 | 88.29 ± 0.83 | 1.73% | 88.99 ± 0.22 | 2.8% | 88.73 ± 0.36 | 4.17% | **89.09 ± 0.56** | 6.01% | 89.01 ± 0.54 | 8.99% |
| 16 | 88.82 ± 0.6 | 1.7% | 89.22 ± 0.13 | 2.79% | 88.9 ± 0.37 | 4.17% | 89.25 ± 0.53 | 6.11% | **89.36 ± 0.5** | 9.03% |
| | | | | | **ImageNet-100** | | | | | |
| 1 | **79.75 ± 1.23** | 0.0% | 79.46 ± 1.33 | 0.0% | 79.54 ± 1.2 | 0.0% | 79.4 ± 1.51 | 0.0% | 79.12 ± 1.07 | 0.0% |
| 2 | 88.0 ± 0.79 | 3.92% | 88.65 ± 1.08 | 6.19% | 88.69 ± 0.87 | 8.61% | 88.88 ± 1.03 | 11.1% | **89.4 ± 0.78** | 14.59% |
| 3 | 89.94 ± 0.77 | 1.04% | 90.09 ± 0.72 | 1.7% | 90.24 ± 0.88 | 2.52% | **90.4 ± 0.75** | 3.63% | 90.29 ± 1.22 | 5.49% |
| 4 | 90.84 ± 0.71 | 0.78% | 90.92 ± 0.63 | 1.29% | **91.52 ± 0.58** | 1.89% | 91.46 ± 0.48 | 2.74% | 91.42 ± 0.53 | 4.19% |
| 5 | 91.66 ± 0.53 | 0.7% | 91.75 ± 0.41 | 1.12% | **92.04 ± 0.6** | 1.62% | 91.92 ± 0.38 | 2.38% | 91.93 ± 0.46 | 3.64% |
| 6 | 92.3 ± 0.46 | 0.65% | 92.2 ± 0.34 | 1.0% | **92.37 ± 0.48** | 1.45% | 92.31 ± 0.29 | 2.14% | 92.18 ± 0.45 | 3.26% |
| 7 | 92.53 ± 0.34 | 0.59% | 92.55 ± 0.36 | 0.91% | **92.71 ± 0.38** | 1.36% | 92.55 ± 0.43 | 2.06% | 92.62 ± 0.4 | 3.16% |
| 8 | 92.64 ± 0.43 | 0.56% | 92.72 ± 0.37 | 0.86% | 92.84 ± 0.29 | 1.29% | 92.87 ± 0.3 | 1.93% | **92.94 ± 0.2** | 2.99% |
| 9 | 92.82 ± 0.41 | 0.54% | 93.02 ± 0.18 | 0.83% | 93.12 ± 0.39 | 1.23% | 93.06 ± 0.23 | 1.91% | **93.14 ± 0.24** | 2.92% |
| 10 | 93.04 ± 0.52 | 0.51% | 93.19 ± 0.31 | 0.83% | 93.15 ± 0.35 | 1.21% | **93.32 ± 0.27** | 1.86% | 93.24 ± 0.35 | 2.9% |
| 11 | 93.07 ± 0.35 | 0.52% | **93.48 ± 0.35** | 0.81% | 93.43 ± 0.22 | 1.23% | 93.46 ± 0.26 | 1.83% | 93.35 ± 0.43 | 2.88% |
| 12 | 93.26 ± 0.37 | 0.49% | 93.54 ± 0.44 | 0.83% | 93.55 ± 0.22 | 1.18% | **93.57 ± 0.17** | 1.74% | 93.54 ± 0.47 | 2.85% |
| 13 | 93.4 ± 0.41 | 0.49% | 93.66 ± 0.41 | 0.79% | 93.7 ± 0.2 | 1.13% | 93.62 ± 0.17 | 1.77% | **93.72 ± 0.45** | 2.75% |
| 14 | 93.38 ± 0.45 | 0.5% | 93.8 ± 0.34 | 0.8% | 93.76 ± 0.33 | 1.12% | **94.04 ± 0.4** | 1.76% | 93.71 ± 0.32 | 2.74% |
| 15 | 93.57 ± 0.53 | 0.5% | 93.83 ± 0.53 | 0.78% | 93.9 ± 0.13 | 1.16% | **94.04 ± 0.37** | 1.77% | 93.93 ± 0.44 | 2.68% |
| 16 | 93.78 ± 0.59 | 0.49% | 94.01 ± 0.6 | 0.76% | 93.9 ± 0.13 | 1.15% | **94.22 ± 0.42** | 1.7% | 94.06 ± 0.32 | 2.62% |
| | | | | | **DomainNet-Real** | | | | | |
| 1 | 68.31 ± 0.39 | 0.0% | **68.34 ± 0.5** | 0.0% | 68.19 ± 0.47 | 0.0% | 68.13 ± 0.52 | 0.0% | 67.96 ± 0.47 | 0.0% |
| 2 | 74.15 ± 0.19 | 3.16% | 74.56 ± 0.47 | 5.09% | 74.15 ± 0.52 | 7.63% | 74.37 ± 0.29 | 10.57% | **74.97 ± 0.09** | 15.42% |
| 3 | 76.4 ± 0.33 | 1.66% | 76.9 ± 0.44 | 2.73% | 76.96 ± 0.31 | 4.11% | **77.19 ± 0.22** | 5.95% | 77.16 ± 0.22 | 9.27% |
| 4 | 77.96 ± 0.51 | 1.57% | 78.31 ± 0.48 | 2.52% | 78.67 ± 0.25 | 3.69% | **78.88 ± 0.26** | 5.24% | 78.68 ± 0.39 | 8.29% |
| 5 | 79.24 ± 0.47 | 1.56% | 79.55 ± 0.38 | 2.51% | 79.85 ± 0.3 | 3.57% | **80.19 ± 0.28** | 5.15% | 79.88 ± 0.21 | 8.08% |
| 6 | 80.13 ± 0.37 | 1.56% | 80.6 ± 0.24 | 2.52% | 80.7 ± 0.26 | 3.55% | **81.11 ± 0.27** | 5.02% | 80.8 ± 0.31 | 8.04% |
| 7 | 80.94 ± 0.4 | 1.58% | 81.35 ± 0.34 | 2.52% | 81.51 ± 0.11 | 3.52% | **81.85 ± 0.19** | 5.07% | 81.58 ± 0.21 | 7.96% |
| 8 | 81.7 ± 0.33 | 1.59% | 82.07 ± 0.2 | 2.52% | 82.27 ± 0.1 | 3.57% | **82.28 ± 0.12** | 5.05% | 82.21 ± 0.13 | 8.05% |
| 9 | 82.25 ± 0.3 | 1.57% | 82.65 ± 0.13 | 2.53% | **82.78 ± 0.13** | 3.53% | 82.73 ± 0.1 | 5.12% | 82.77 ± 0.14 | 8.1% |
| 10 | 82.59 ± 0.28 | 1.56% | 83.1 ± 0.17 | 2.55% | 83.16 ± 0.14 | 3.49% | 83.19 ± 0.15 | 5.14% | **83.27 ± 0.16** | 8.15% |
| 11 | 83.05 ± 0.28 | 1.58% | 83.43 ± 0.18 | 2.54% | 83.6 ± 0.19 | 3.52% | 83.53 ± 0.07 | 5.19% | **83.65 ± 0.19** | 8.38% |
| 12 | 81.0 ± 0.32 | 1.53% | 81.33 ± 0.47 | 2.55% | 81.88 ± 0.32 | 3.48% | 82.43 ± 0.2 | 5.19% | **82.79 ± 0.27** | 8.14% |
| 13 | 83.27 ± 0.26 | 2.89% | 83.27 ± 0.17 | 5.16% | 83.5 ± 0.22 | 6.59% | 83.48 ± 0.17 | 8.36% | **83.75 ± 0.11** | 13.12% |
| 14 | 83.8 ± 0.18 | 1.88% | 84.03 ± 0.04 | 3.28% | 84.03 ± 0.16 | 4.89% | 84.03 ± 0.15 | 7.55% | **84.12 ± 0.2** | 10.59% |
| 15 | 84.15 ± 0.1 | 1.71% | 84.35 ± 0.13 | 2.86% | 84.3 ± 0.12 | 4.42% | 84.32 ± 0.13 | 6.68% | **84.39 ± 0.27** | 9.86% |
| 16 | 84.36 ± 0.12 | 1.64% | 84.56 ± 0.17 | 2.73% | 84.58 ± 0.15 | 4.32% | 84.59 ± 0.24 | 6.33% | **84.72 ± 0.17** | 9.33% |

Table 2: Impact of the number of dropout iterations. We evaluate 4 different settings of $M$ on 4 natural image datasets - CIFAR100, Food101, ImageNet-100, and DomainNet-Real. We report the mean accuracy averaged over 5 runs along with the standard deviation at each AL iteration. **Bold** values represent the **first place** mean accuracy at iteration $t$ with the second place value underlined. We also report the corresponding fraction of the unlabeled examples added to the candidate set.

| $t$ | 3 Mean ± Std | Fraction | 5 Mean ± Std | Fraction | 7 Mean ± Std | Fraction | 9 Mean ± Std | Fraction |
|---|---|---|---|---|---|---|---|---|
| | | | | CIFAR100 | | | | |
| 1 | **72.58 ± 2.18** | 0.0% | 72.58 ± 2.18 | 0.0% | 72.58 ± 2.18 | 0.0% | 72.58 ± 2.18 | 0.0% |
| 2 | 83.52 ± 0.76 | 21.53% | **84.16 ± 1.31** | 21.64% | 83.46 ± 1.04 | 21.7% | 83.55 ± 0.88 | 21.72% |
| 3 | **86.6 ± 0.78** | 11.7% | 85.9 ± 0.94 | 11.37% | 85.82 ± 0.69 | 11.54% | 86.2 ± 0.55 | 11.35% |
| 4 | 87.67 ± 0.57 | 9.22% | **87.76 ± 0.46** | 9.12% | 87.39 ± 1.02 | 8.81% | 87.6 ± 0.37 | 8.78% |
| 5 | **88.94 ± 0.31** | 8.33% | 88.72 ± 0.38 | 7.89% | 88.21 ± 0.73 | 7.62% | 88.43 ± 0.48 | 7.65% |
| 6 | **89.35 ± 0.16** | 7.72% | 89.17 ± 0.24 | 7.26% | 89.06 ± 0.42 | 7.17% | 88.98 ± 0.33 | 7.01% |
| 7 | **89.65 ± 0.31** | 7.37% | 89.53 ± 0.2 | 6.97% | 89.53 ± 0.33 | 6.7% | 89.6 ± 0.18 | 6.69% |
| 8 | **90.12 ± 0.14** | 7.07% | 89.71 ± 0.21 | 6.62% | 90.07 ± 0.26 | 6.42% | 89.86 ± 0.18 | 6.3% |
| 9 | **90.23 ± 0.21** | 6.88% | 90.11 ± 0.38 | 6.55% | 89.99 ± 0.25 | 6.23% | 90.02 ± 0.21 | 6.15% |
| 10 | **90.48 ± 0.22** | 6.75% | 90.39 ± 0.27 | 6.36% | 90.41 ± 0.37 | 6.09% | 90.26 ± 0.11 | 6.02% |
| 11 | **90.6 ± 0.28** | 6.68% | 90.5 ± 0.13 | 6.27% | 90.59 ± 0.36 | 5.98% | 90.52 ± 0.19 | 5.84% |
| 12 | **90.8 ± 0.27** | 6.54% | 90.59 ± 0.24 | 6.24% | 90.63 ± 0.34 | 5.86% | 90.65 ± 0.17 | 5.63% |
| 13 | **90.99 ± 0.2** | 6.37% | 90.81 ± 0.14 | 6.02% | 90.96 ± 0.27 | 5.82% | 90.84 ± 0.21 | 5.64% |
| 14 | **91.26 ± 0.25** | 6.35% | 90.92 ± 0.17 | 6.02% | 91.05 ± 0.22 | 5.7% | 91.01 ± 0.18 | 5.59% |
| 15 | **91.32 ± 0.26** | 6.27% | 91.05 ± 0.06 | 5.85% | 91.13 ± 0.33 | 5.59% | 91.01 ± 0.12 | 5.31% |
| 16 | **91.47 ± 0.16** | 6.0% | 91.21 ± 0.15 | 5.73% | 91.37 ± 0.19 | 5.42% | 91.19 ± 0.21 | 5.21% |
| | | | | Food101 | | | | |
| 1 | **71.33 ± 1.38** | 0.0% | 71.33 ± 1.38 | 0.0% | 71.33 ± 1.38 | 0.0% | 71.33 ± 1.38 | 0.0% |
| 2 | **76.09 ± 1.34** | 8.49% | 75.65 ± 1.37 | 7.81% | 75.74 ± 1.74 | 7.43% | 75.62 ± 2.14 | 7.22% |
| 3 | **79.76 ± 0.88** | 8.6% | 79.15 ± 1.48 | 8.16% | 79.73 ± 1.55 | 7.72% | 78.55 ± 2.43 | 7.36% |
| 4 | **82.24 ± 0.52** | 8.75% | 81.6 ± 0.86 | 8.05% | 81.88 ± 1.06 | 7.44% | 81.27 ± 1.39 | 7.41% |
| 5 | **83.75 ± 0.84** | 8.92% | 83.01 ± 0.8 | 8.23% | 83.32 ± 1.18 | 7.64% | 82.87 ± 1.08 | 7.44% |
| 6 | 84.71 ± 0.58 | 8.58% | **84.73 ± 0.78** | 8.22% | 84.33 ± 1.07 | 7.59% | 84.2 ± 1.06 | 7.51% |
| 7 | **85.72 ± 0.57** | 8.79% | 85.3 ± 0.68 | 8.15% | 85.56 ± 0.88 | 7.69% | 85.05 ± 0.88 | 7.47% |
| 8 | **86.31 ± 0.4** | 8.75% | 86.08 ± 0.49 | 8.09% | 86.3 ± 0.59 | 7.95% | 85.72 ± 0.67 | 7.44% |
| 9 | 86.87 ± 0.29 | 8.84% | **87.04 ± 0.44** | 8.49% | 86.77 ± 0.69 | 7.97% | 86.3 ± 0.7 | 7.73% |
| 10 | 87.27 ± 0.45 | 8.96% | **87.41 ± 0.75** | 8.28% | 87.31 ± 0.64 | 8.02% | 87.03 ± 0.76 | 7.72% |
| 11 | 87.72 ± 0.55 | 8.8% | **87.8 ± 0.88** | 8.47% | 87.63 ± 0.66 | 7.88% | 87.55 ± 0.68 | 7.8% |
| 12 | 88.07 ± 0.63 | 8.77% | 88.06 ± 0.72 | 8.85% | **88.1 ± 0.64** | 7.92% | 87.84 ± 0.77 | 7.86% |
| 13 | **88.51 ± 0.59** | 8.9% | 88.44 ± 0.51 | 8.58% | 88.51 ± 0.8 | 8.03% | 88.11 ± 0.61 | 7.75% |
| 14 | **88.92 ± 0.67** | 8.94% | 88.59 ± 0.73 | 8.48% | 88.86 ± 0.74 | 7.95% | 88.61 ± 0.64 | 7.97% |
| 15 | 89.01 ± 0.54 | 8.99% | 88.9 ± 0.55 | 8.55% | **89.04 ± 0.85** | 7.99% | 88.89 ± 0.77 | 7.86% |
| 16 | **89.36 ± 0.5** | 9.03% | 89.28 ± 0.7 | 8.39% | 89.29 ± 0.73 | 7.92% | 89.29 ± 0.73 | 7.82% |
| | | | | ImageNet-100 | | | | |
| 1 | **79.12 ± 1.07** | 0.0% | 79.12 ± 1.07 | 0.0% | 79.12 ± 1.07 | 0.0% | 79.12 ± 1.07 | 0.0% |
| 2 | **89.4 ± 0.78** | 14.59% | 89.38 ± 1.01 | 14.78% | 89.06 ± 1.07 | 14.87% | 88.79 ± 1.1 | 14.92% |
| 3 | 90.29 ± 1.22 | 5.49% | 90.48 ± 0.79 | 5.37% | **90.49 ± 1.18** | 5.26% | 90.23 ± 0.82 | 5.24% |
| 4 | 91.42 ± 0.53 | 4.19% | **91.47 ± 0.42** | 4.02% | 91.24 ± 0.63 | 3.93% | 91.14 ± 0.54 | 3.86% |
| 5 | **91.93 ± 0.46** | 3.64% | 91.89 ± 0.43 | 3.44% | 91.9 ± 0.36 | 3.36% | 91.84 ± 0.56 | 3.31% |
| 6 | 92.18 ± 0.45 | 3.26% | 92.43 ± 0.56 | 3.08% | 92.17 ± 0.48 | 2.96% | **92.47 ± 0.61** | 2.96% |
| 7 | 92.62 ± 0.4 | 3.16% | 92.68 ± 0.58 | 2.85% | 92.67 ± 0.4 | 2.76% | **92.72 ± 0.57** | 2.75% |
| 8 | 92.94 ± 0.2 | 2.99% | 92.79 ± 0.61 | 2.74% | 92.72 ± 0.38 | 2.63% | **92.95 ± 0.26** | 2.6% |
| 9 | 93.14 ± 0.24 | 2.92% | **93.2 ± 0.33** | 2.68% | 92.91 ± 0.26 | 2.58% | 92.97 ± 0.31 | 2.51% |
| 10 | 93.24 ± 0.35 | 2.9% | **93.37 ± 0.21** | 2.6% | 93.09 ± 0.18 | 2.53% | 93.14 ± 0.31 | 2.45% |
| 11 | 93.35 ± 0.43 | 2.88% | 93.38 ± 0.22 | 2.54% | **93.39 ± 0.17** | 2.42% | 93.34 ± 0.18 | 2.37% |
| 12 | **93.54 ± 0.47** | 2.85% | 93.5 ± 0.22 | 2.53% | 93.42 ± 0.18 | 2.35% | 93.41 ± 0.21 | 2.29% |
| 13 | **93.72 ± 0.45** | 2.75% | 93.51 ± 0.22 | 2.48% | 93.47 ± 0.2 | 2.31% | 93.38 ± 0.14 | 2.26% |
| 14 | **93.71 ± 0.32** | 2.74% | 93.65 ± 0.17 | 2.43% | 93.56 ± 0.14 | 2.23% | 93.54 ± 0.2 | 2.23% |
| 15 | **93.93 ± 0.44** | 2.68% | 93.65 ± 0.14 | 2.39% | 93.62 ± 0.12 | 2.22% | 93.79 ± 0.27 | 2.18% |
| 16 | **94.06 ± 0.32** | 2.62% | 93.69 ± 0.2 | 2.34% | 93.63 ± 0.24 | 2.23% | 93.83 ± 0.18 | 2.16% |
| | | | | DomainNet-Real | | | | |
| 1 | **67.96 ± 0.47** | 0.0% | 67.96 ± 0.47 | 0.0% | 67.96 ± 0.47 | 0.0% | 67.96 ± 0.47 | 0.0% |
| 2 | 74.97 ± 0.09 | 15.42% | **74.97 ± 0.24** | 15.36% | 74.85 ± 0.31 | 15.29% | 74.94 ± 0.38 | 15.26% |
| 3 | **77.16 ± 0.22** | 9.27% | 77.16 ± 0.26 | 8.94% | 77.06 ± 0.16 | 8.84% | 77.09 ± 0.21 | 8.63% |
| 4 | 78.68 ± 0.39 | 8.29% | 78.67 ± 0.15 | 7.89% | **78.74 ± 0.35** | 7.63% | 78.71 ± 0.26 | 7.48% |
| 5 | 79.88 ± 0.21 | 8.08% | 79.85 ± 0.2 | 7.56% | 79.88 ± 0.19 | 7.36% | **79.95 ± 0.21** | 7.17% |
| 6 | 80.8 ± 0.31 | 8.04% | 80.81 ± 0.17 | 7.55% | **80.86 ± 0.22** | 7.23% | 80.82 ± 0.16 | 7.04% |
| 7 | **81.58 ± 0.21** | 7.96% | 81.47 ± 0.21 | 7.52% | 81.58 ± 0.34 | 7.25% | 81.5 ± 0.19 | 7.0% |
| 8 | 82.21 ± 0.13 | 8.05% | 82.18 ± 0.3 | 7.68% | **82.3 ± 0.24** | 7.22% | 82.07 ± 0.24 | 7.1% |
| 9 | 82.77 ± 0.14 | 8.1% | 82.79 ± 0.28 | 7.65% | **82.8 ± 0.27** | 7.22% | 82.59 ± 0.22 | 7.19% |
| 10 | **83.27 ± 0.16** | 8.15% | 83.19 ± 0.25 | 7.71% | 83.2 ± 0.3 | 7.39% | 83.07 ± 0.33 | 7.27% |
| 11 | **83.65 ± 0.19** | 8.38% | 83.55 ± 0.15 | 7.83% | 83.53 ± 0.17 | 7.44% | 83.53 ± 0.28 | 7.11% |
| 12 | 82.79 ± 0.27 | 8.14% | **82.79 ± 0.24** | 7.56% | 82.49 ± 0.21 | 7.39% | 82.6 ± 0.2 | 7.22% |
| 13 | **83.75 ± 0.11** | 13.12% | 83.65 ± 0.23 | 12.41% | 83.73 ± 0.16 | 12.19% | 83.65 ± 0.15 | 11.76% |
| 14 | 84.12 ± 0.2 | 10.59% | 84.12 ± 0.24 | 9.88% | **84.31 ± 0.11** | 9.85% | 84.22 ± 0.07 | 9.75% |
| 15 | 84.39 ± 0.27 | 9.86% | 84.44 ± 0.19 | 9.1% | **84.6 ± 0.1** | 8.78% | 84.47 ± 0.1 | 8.64% |
| 16 | 84.72 ± 0.17 | 9.33% | 84.67 ± 0.17 | 8.65% | **84.81 ± 0.11** | 8.45% | 84.76 ± 0.04 | 8.28% |

Table 3: Descriptions of the biomedical datasets utilized in our study, which span a wide range of imaging modalities, biomedical disciplines, and task difficulties. Abbreviations: binary classification (BC), hematoxylin and eosin (H&E), immunofluorescence (IF), multiclass classification (MC). All train/ val/ test splits are 0.7/ 0.1/ 0.2 of the total unless noted otherwise in the text.

| Name | Modality | Domain | Stain | Task (Classes) | Image size | Total |
|---|---|---|---|---|---|---|
| Blood Smear Acevedo et al. (2020) | Brightfield | Cytology | Giemsa | MC (8) | (360, 363) | 17092 |
| Colorectal histology Kather et al. (2016) | Brightfield | Pathology | H&E | MC (8) | (150, 150) | 5000 |
| Diabetic Retinopathy Kaggle & EyePacs (2015) | Fundoscopy | Ophthalmology | - | MC (5) | (256, 256) | 2750 |
| IICBU 2008 HeLaMurphy et al. (2000) | Fluorescence | Cell Biology | IF | MC (10) | (382, 382) | 862 |
| Skin cancer Tschandl et al. (2018) | Dermoscopy | Dermatology | - | MC (7) | (450, 600) | 10015 |
| Patch Camelyon Veeling et al. (2018)Ehteshami Bejnordi et al. (2017) | Brightfield | Pathology | H&E | BC (2) | (96, 96) | 327,680 |

Table 4: Test set accuracy with standard deviations (average of 5 random seeds) at certain iterations $t$ during AL on CIFAR100 Krizhevsky (2009), Food101 Bossard et al. (2014), ImageNet-100 Gansbeke et al. (2020), and DomainNet-Real Peng et al. (2019) (from top to bottom) with DINOv2 ViT-g14 as the feature extractor $f$. Typiclust, ProbCover, and our proposed **DropQuery** use their own respective initialization strategies for the cold-start. Cells are color coded according to the magnitude of improvement in mean accuracy over the Random baseline. Green cells are positive with better performance than random whereas red cells are negative with worse performance than random. **Bold** values represent the **first place** mean accuracy at iteration $t$ with the second place value underlined.

| $t$ | Random | Uncertainty | Entropy | Margins | BALD | pBALD | Coreset | Typiclust | ProbCover | BADGE | Alfamix | DropQuery (Ours) |
|---|---|---|---|---|---|---|---|---|---|---|---|---|
| | | | | | | CIFAR100 | | | | | | |
| 1 | 48.1±2.1 | 48.1±2.1 | 48.1±2.1 | 48.1±2.1 | 48.1±2.1 | 48.1±2.1 | 48.1±2.1 | 64.6±2.0 | 62.3±1.1 | 48.1±2.1 | 48.1±2.1 | **72.6±2.2** |
| 2 | 64.5±2.7 | 62.6±1.8 | 58.0±2.3 | 69.9±2.6 | 67.4±1.7 | 71.2±1.3 | 64.5±2.2 | 80.8±0.2 | 76.2±1.9 | 72.5±2.0 | 78.0±1.1 | **83.5±0.8** |
| 4 | 78.7±1.6 | 74.6±3.4 | 70.4±3.2 | 82.6±0.8 | 80.4±1.6 | 83.9±1.3 | 78.2±1.5 | 86.8±0.4 | 81.9±0.8 | 84.1±0.5 | 83.9±1.6 | **87.7±0.6** |
| 6 | 83.3±0.6 | 81.1±2.5 | 76.1±3.6 | 86.0±0.6 | 83.7±0.8 | 87.2±0.4 | 82.2±1.1 | 87.7±0.4 | 84.9±0.9 | 87.2±0.3 | 85.8±1.8 | **89.3±0.2** |
| 8 | 86.1±0.4 | 84.2±0.9 | 81.4±2.9 | 87.9±0.8 | 85.2±0.6 | 88.4±0.6 | 84.3±1.0 | 88.4±0.4 | 86.5±0.8 | 88.8±0.1 | 87.8±1.4 | **90.1±0.1** |
| 10 | 87.3±0.5 | 85.9±0.8 | 84.2±1.9 | 88.9±0.4 | 86.3±1.1 | 89.5±0.3 | 85.8±0.7 | 88.7±0.1 | 87.3±0.6 | 89.5±0.2 | 89.2±0.7 | **90.5±0.2** |
| 12 | 88.2±0.3 | 87.1±0.5 | 85.6±1.5 | 89.7±0.2 | 87.1±1.0 | 89.9±0.2 | 86.8±0.9 | 89.0±0.3 | 88.2±0.6 | 90.1±0.1 | 89.8±0.5 | **90.8±0.3** |
| 14 | 88.9±0.3 | 88.3±0.5 | 86.9±1.3 | 90.2±0.4 | 87.8±0.9 | 90.4±0.1 | 87.5±0.7 | 89.2±0.1 | 88.7±0.5 | 90.6±0.2 | 90.3±0.7 | **91.3±0.3** |
| 16 | 89.2±0.3 | 89.2±0.3 | 87.8±0.9 | 90.6±0.4 | 88.4±0.7 | 90.8±0.1 | 88.3±0.8 | 89.3±0.2 | 89.1±0.3 | 90.9±0.1 | 90.8±0.5 | **91.5±0.2** |
| 18 | 89.5±0.3 | 89.8±0.3 | 88.7±0.9 | 91.0±0.1 | 88.9±0.8 | 91.1±0.2 | 88.7±0.9 | 89.5±0.3 | 89.5±0.1 | 91.2±0.1 | 91.1±0.4 | **91.6±0.2** |
| 20 | 89.8±0.2 | 90.3±0.4 | 89.3±0.6 | 91.3±0.1 | 89.4±0.7 | 91.3±0.1 | 89.0±0.6 | 89.5±0.1 | 89.9±0.2 | 91.4±0.1 | 91.3±0.3 | **91.9±0.3** |
| | | | | | | Food101 | | | | | | |
| 1 | 47.4±2.0 | 47.4±2.0 | 47.4±2.0 | 47.4±2.0 | 47.4±2.0 | 47.4±2.0 | 47.4±2.0 | 68.3±1.8 | 66.1±2.0 | 47.4±2.0 | 47.4±2.0 | **71.3±1.4** |
| 2 | 64.4±2.0 | 50.9±3.4 | 49.9±3.0 | 62.7±3.2 | 50.8±3.3 | 67.1±2.6 | 51.5±2.7 | 79.1±0.6 | 72.3±1.2 | 64.8±1.5 | 74.1±0.9 | 76.1±1.3 |
| 4 | 77.2±1.3 | 63.5±4.1 | 58.5±3.7 | 75.5±3.0 | 62.2±2.5 | 78.2±2.6 | 63.0±2.5 | 83.1±0.7 | 78.1±1.6 | 77.5±1.9 | 79.6±0.8 | 82.2±0.5 |
| 6 | 81.4±0.7 | 72.3±2.8 | 66.9±3.6 | 80.7±1.8 | 69.3±2.1 | 83.0±0.8 | 69.9±2.7 | 84.9±0.5 | 80.3±1.5 | 82.1±0.9 | 83.2±1.2 | 84.7±0.6 |
| 8 | 83.8±0.2 | 77.3±2.5 | 73.0±3.6 | 84.7±1.0 | 74.2±1.8 | 85.6±0.5 | 73.4±2.2 | 86.0±0.6 | 81.9±1.2 | 85.3±1.1 | 85.6±0.8 | 86.3±0.4 |
| 10 | 85.3±0.2 | 80.5±2.3 | 76.8±3.0 | 86.6±0.9 | 77.4±0.9 | 87.4±0.5 | 75.6±2.2 | 86.6±0.3 | 83.4±1.0 | 86.9±0.9 | 86.7±0.9 | 87.3±0.5 |
| 12 | 86.4±0.2 | 81.9±1.3 | 79.8±2.4 | 87.6±0.6 | 79.1±1.2 | 88.2±0.4 | 77.6±2.0 | 86.9±0.2 | 83.8±1.2 | 87.8±0.8 | 87.9±0.4 | 88.1±0.6 |
| 14 | 86.9±0.6 | 83.8±0.8 | 81.2±1.7 | 88.9±0.5 | 81.2±0.7 | 88.9±0.4 | 79.7±0.8 | 87.1±0.3 | 84.5±0.7 | 88.2±0.8 | 88.6±0.3 | 88.9±0.7 |
| 16 | 87.5±0.2 | 85.2±0.8 | 82.7±1.2 | **89.4±0.3** | 82.3±0.5 | 89.4±0.3 | 80.8±1.3 | 87.3±0.2 | 85.1±0.4 | 89.4±0.5 | 89.2±0.2 | 89.4±0.5 |
| 18 | 88.0±0.3 | 86.5±0.4 | 83.9±1.1 | 89.8±0.4 | 83.0±0.8 | 89.8±0.2 | 82.0±1.4 | 87.5±0.2 | 85.3±0.7 | 89.9±0.5 | 89.7±0.2 | 89.8±0.4 |
| 20 | 88.3±0.4 | 87.3±0.2 | 84.6±1.2 | 90.3±0.3 | 83.9±0.7 | 90.1±0.1 | 83.0±1.0 | 87.8±0.2 | 85.5±0.6 | 90.2±0.5 | 90.0±0.2 | **90.4±0.1** |
| | | | | | | ImageNet-100 | | | | | | |
| 1 | 54.7±2.6 | 54.7±2.6 | 54.7±2.6 | 54.7±2.6 | 54.7±2.6 | 54.7±2.6 | 54.7±2.6 | 76.7±1.7 | 76.6±1.7 | 54.7±2.6 | 54.7±2.6 | **79.1±1.1** |
| 2 | 76.5±3.4 | 60.0±7.0 | 57.8±6.1 | 79.0±1.8 | 78.7±4.1 | 83.9±2.4 | 76.5±3.3 | 89.5±0.2 | 89.1±1.3 | 81.8±1.4 | 84.9±1.9 | 89.4±0.8 |
| 4 | 86.8±1.0 | 74.4±6.1 | 68.6±8.0 | 89.0±0.6 | 88.2±0.9 | 90.6±0.7 | 85.7±2.2 | 92.3±0.2 | 91.7±0.2 | 89.8±0.8 | 90.7±1.0 | 91.5±0.5 |
| 6 | 89.7±0.7 | 82.5±1.4 | 78.6±4.1 | 90.7±0.6 | 89.4±0.2 | 92.0±0.7 | 88.0±1.2 | 93.1±0.2 | 92.4±0.6 | 91.7±0.5 | 92.4±0.6 | 92.3±0.4 |
| 8 | 91.0±0.6 | 85.7±1.4 | 81.7±3.5 | 92.2±0.3 | 89.9±0.3 | 92.9±0.4 | 89.0±0.7 | 93.3±0.4 | 92.7±0.4 | 92.6±0.9 | 93.4±0.3 | 93.0±0.1 |
| 10 | 91.8±0.8 | 88.1±1.1 | 85.1±2.0 | 92.9±0.3 | 90.2±0.4 | 93.7±0.4 | 89.8±0.4 | 93.4±0.3 | 93.0±0.1 | 93.1±0.5 | 93.8±0.4 | 93.3±0.3 |
| 12 | 92.4±0.6 | 89.2±0.6 | 87.6±1.3 | 93.5±0.2 | 90.5±0.5 | 93.9±0.3 | 90.4±0.5 | 93.4±0.3 | 93.0±0.3 | 93.6±0.4 | 94.1±0.3 | 93.6±0.4 |
| 14 | 92.9±0.6 | 90.2±0.5 | 88.8±1.1 | 93.9±0.2 | 91.0±0.5 | 94.2±0.2 | 90.9±0.4 | 93.5±0.3 | 93.3±0.2 | 94.1±0.4 | 94.1±0.3 | 93.7±0.3 |
| 16 | 93.3±0.5 | 90.5±0.6 | 89.7±0.9 | 94.2±0.3 | 91.4±0.2 | 94.4±0.3 | 91.1±0.2 | 93.4±0.3 | 93.3±0.2 | 94.1±0.4 | 94.4±0.2 | 94.1±0.3 |
| 18 | 93.3±0.4 | 91.0±0.6 | 90.4±0.9 | 94.3±0.3 | 91.8±0.4 | 94.4±0.4 | 91.4±0.6 | 93.5±0.4 | 93.5±0.2 | 94.3±0.4 | 94.7±0.3 | 94.1±0.3 |
| 20 | 93.4±0.5 | 92.0±0.6 | 90.8±0.7 | 94.4±0.3 | 92.1±0.3 | 94.5±0.2 | 91.8±0.4 | 93.6±0.3 | 93.7±0.1 | 94.6±0.4 | **94.8±0.3** | 94.4±0.4 |
| | | | | | | DomainNet-Real | | | | | | |
| 1 | 44.8±0.8 | 44.8±0.8 | 44.8±0.8 | 44.8±0.8 | 44.8±0.8 | 44.8±0.8 | 44.8±0.8 | 64.8±0.5 | 63.9±0.4 | 44.8±0.8 | 44.8±0.8 | **68.0±0.5** |
| 2 | 61.8±0.9 | 54.1±1.1 | 50.4±1.3 | 61.9±0.9 | 60.3±1.0 | 65.3±1.4 | 58.8±1.3 | 73.0±0.7 | 73.8±0.2 | 65.2±1.5 | 71.4±0.8 | **75.0±0.1** |
| 4 | 73.1±0.7 | 64.5±1.0 | 58.7±3.2 | 72.5±0.4 | 69.3±0.5 | 75.6±0.3 | 69.2±1.0 | 74.8±0.6 | 77.8±0.1 | 75.4±0.8 | 76.6±0.3 | **78.7±0.4** |
| 6 | 77.0±0.4 | 70.6±0.4 | 65.7±1.9 | 77.2±0.6 | 73.7±0.3 | 79.1±0.3 | 73.8±0.4 | 75.7±0.4 | 79.7±0.4 | 78.6±0.9 | 77.8±0.2 | **80.8±0.3** |
| 8 | 79.2±0.2 | 74.1±0.5 | 70.3±1.7 | 79.9±0.3 | 76.4±0.1 | 80.7±0.3 | 76.0±0.3 | 76.2±0.4 | 80.6±0.1 | 80.7±0.5 | 78.7±0.2 | **82.2±0.1** |
| 10 | 80.4±0.2 | 76.5±0.4 | 73.8±1.2 | 81.6±0.3 | 78.0±0.3 | 82.0±0.3 | 77.7±0.5 | 76.8±0.3 | 81.3±0.1 | 82.2±0.3 | 79.0±0.2 | **83.3±0.2** |
| 12 | 80.3±0.1 | 77.0±0.3 | 75.5±0.6 | 81.6±0.3 | 78.4±0.5 | 82.0±0.2 | 78.1±0.6 | 77.0±0.1 | 80.9±0.3 | 81.9±0.4 | 78.7±0.3 | **82.8±0.3** |
| 14 | 81.4±0.3 | 79.1±0.4 | 77.5±0.4 | 83.0±0.2 | 79.8±0.3 | 83.2±0.3 | 79.5±0.6 | 77.6±0.3 | 81.3±0.1 | 83.6±0.2 | 79.4±0.2 | **84.1±0.2** |
| 16 | 82.1±0.3 | 80.4±0.3 | 79.0±0.6 | 84.0±0.3 | 81.1±0.2 | 84.0±0.1 | 80.4±0.6 | 78.0±0.5 | 82.3±0.1 | 84.3±0.3 | 79.7±0.3 | **84.7±0.2** |
| 18 | 82.5±0.2 | 81.5±0.3 | 80.0±0.4 | 84.5±0.2 | 81.6±0.2 | 84.6±0.3 | 81.2±0.6 | 78.5±0.2 | 83.0±0.2 | 84.9±0.2 | 79.9±0.1 | **85.1±0.0** |
| 20 | 83.0±0.2 | 82.2±0.2 | 81.0±0.5 | 85.0±0.3 | 82.3±0.3 | 84.9±0.3 | 81.7±0.6 | 78.7±0.4 | 83.4±0.1 | 85.3±0.2 | 80.2±0.2 | **85.5±0.1** |

Table 5: Test set accuracy with standard deviations (average of 5 random seeds) at certain iterations $t$ during AL on other VTAB+ Zhai et al. (2020) datasets including Stanford Cars Krause et al. (2013), FVGC Aircraft Maji et al. (2013), Oxford-IIIT Pets Parkhi et al. (2012), and the large Places365 Zhou et al. (2017) dataset (from top to bottom) with OpenCLIP ViT-G14 as the feature extractor $f$. Cells are color coded according to the magnitude of improvement in mean accuracy over the Random baseline. Probcover was intractable on Places365 with the resources available, and thus not performed (-). **Bold** values represent the **first place** mean accuracy at iteration $t$ with the second place value underlined.

| $t$ | Random | Uncertainty | Entropy | Margins | BALD | pBALD | Coreset | Typiclust | ProbCover | BADGE | Alfamix | DropQuery (Ours) |
|---|---|---|---|---|---|---|---|---|---|---|---|---|
| | | | | | | Stanford Cars | | | | | | |
| 1 | $51.9 \pm 1.1$ | $51.9 \pm 1.1$ | $51.9 \pm 1.1$ | $51.9 \pm 1.1$ | $52.1 \pm 1.2$ | $51.9 \pm 1.1$ | $51.9 \pm 1.1$ | $\mathbf{74.2 \pm 1.6}$ | $\underline{74.0 \pm 2.2}$ | $51.9 \pm 1.1$ | $51.9 \pm 1.1$ | $73.4 \pm 1.7$ |
| 2 | $72.3 \pm 1.7$ | $69.0 \pm 1.1$ | $66.3 \pm 1.5$ | $75.7 \pm 2.1$ | $69.2 \pm 1.6$ | $76.6 \pm 1.5$ | $76.1 \pm 0.4$ | $\mathbf{87.9 \pm 0.6}$ | $86.1 \pm 1.9$ | $77.2 \pm 1.6$ | $82.4 \pm 0.6$ | $\underline{87.8 \pm 0.5}$ |
| 4 | $87.0 \pm 1.1$ | $87.1 \pm 1.1$ | $84.1 \pm 1.4$ | $90.4 \pm 0.9$ | $86.1 \pm 1.2$ | $90.7 \pm 0.6$ | $87.8 \pm 0.8$ | $91.3 \pm 0.4$ | $91.2 \pm 0.2$ | $90.4 \pm 0.7$ | $90.4 \pm 0.2$ | $\mathbf{92.9 \pm 0.3}$ |
| 6 | $91.2 \pm 0.2$ | $92.8 \pm 0.3$ | $90.9 \pm 0.5$ | $93.8 \pm 0.3$ | $91.3 \pm 0.4$ | $93.2 \pm 0.6$ | $90.8 \pm 0.4$ | $92.1 \pm 0.3$ | $92.7 \pm 0.2$ | $\underline{93.9 \pm 0.1}$ | $92.0 \pm 0.3$ | $\mathbf{94.4 \pm 0.3}$ |
| 8 | $93.2 \pm 0.3$ | $94.6 \pm 0.2$ | $94.2 \pm 0.6$ | $95.0 \pm 0.2$ | $94.2 \pm 0.3$ | $94.5 \pm 0.4$ | $92.3 \pm 0.3$ | $92.6 \pm 0.5$ | $93.4 \pm 0.1$ | $\underline{95.0 \pm 0.3}$ | $92.8 \pm 0.4$ | $\mathbf{95.1 \pm 0.1}$ |
| 10 | $94.0 \pm 0.4$ | $95.4 \pm 0.2$ | $\underline{95.5 \pm 0.3}$ | $95.5 \pm 0.2$ | $95.3 \pm 0.2$ | $95.2 \pm 0.2$ | $93.4 \pm 0.3$ | $93.1 \pm 0.5$ | $93.9 \pm 0.2$ | $\mathbf{95.6 \pm 0.2}$ | $93.3 \pm 0.3$ | $95.4 \pm 0.1$ |
| 12 | $94.4 \pm 0.2$ | $95.7 \pm 0.1$ | $\underline{95.8 \pm 0.1}$ | $95.7 \pm 0.2$ | $95.8 \pm 0.2$ | $95.5 \pm 0.2$ | $94.2 \pm 0.3$ | $93.6 \pm 0.3$ | $94.2 \pm 0.1$ | $95.8 \pm 0.2$ | $93.9 \pm 0.3$ | $\mathbf{95.8 \pm 0.1}$ |
| 14 | $94.6 \pm 0.2$ | $95.8 \pm 0.1$ | $\underline{95.9 \pm 0.1}$ | $95.8 \pm 0.2$ | $95.9 \pm 0.0$ | $95.7 \pm 0.1$ | $94.5 \pm 0.3$ | $93.7 \pm 0.4$ | $94.7 \pm 0.1$ | $95.8 \pm 0.2$ | $94.5 \pm 0.3$ | $\mathbf{96.0 \pm 0.1}$ |
| 16 | $95.0 \pm 0.3$ | $\underline{95.9 \pm 0.1}$ | $95.9 \pm 0.1$ | $\mathbf{96.0 \pm 0.1}$ | $95.9 \pm 0.1$ | $95.8 \pm 0.1$ | $94.7 \pm 0.3$ | $94.0 \pm 0.4$ | $95.0 \pm 0.2$ | $95.8 \pm 0.2$ | $94.7 \pm 0.2$ | $95.9 \pm 0.1$ |
| 18 | $95.1 \pm 0.3$ | $95.9 \pm 0.1$ | $95.9 \pm 0.1$ | $\mathbf{95.9 \pm 0.2}$ | $95.8 \pm 0.2$ | $95.9 \pm 0.1$ | $94.9 \pm 0.1$ | $94.1 \pm 0.3$ | $95.2 \pm 0.2$ | $95.9 \pm 0.1$ | $94.9 \pm 0.3$ | $\underline{95.9 \pm 0.1}$ |
| 20 | $95.2 \pm 0.2$ | $95.9 \pm 0.0$ | $\underline{95.9 \pm 0.1}$ | $95.9 \pm 0.1$ | $\mathbf{96.0 \pm 0.0}$ | $95.9 \pm 0.1$ | $95.2 \pm 0.2$ | $94.2 \pm 0.4$ | $95.2 \pm 0.1$ | $95.9 \pm 0.1$ | $95.1 \pm 0.0$ | $95.9 \pm 0.1$ |
| | | | | | | FVGC Aircraft | | | | | | |
| 1 | $31.6 \pm 1.6$ | $31.6 \pm 1.6$ | $31.6 \pm 1.6$ | $31.6 \pm 1.6$ | $31.6 \pm 1.8$ | $31.6 \pm 1.6$ | $31.6 \pm 1.6$ | $\underline{42.0 \pm 2.0}$ | $34.3 \pm 0.3$ | $31.6 \pm 1.6$ | $31.6 \pm 1.6$ | $\mathbf{42.5 \pm 1.3}$ |
| 2 | $42.7 \pm 0.6$ | $39.5 \pm 1.6$ | $37.6 \pm 2.4$ | $42.9 \pm 1.7$ | $39.2 \pm 2.5$ | $44.1 \pm 1.7$ | $39.2 \pm 1.2$ | $\underline{51.9 \pm 0.7}$ | $41.8 \pm 0.7$ | $44.5 \pm 2.8$ | $48.5 \pm 1.0$ | $\mathbf{52.2 \pm 0.6}$ |
| 4 | $53.2 \pm 1.3$ | $50.8 \pm 1.0$ | $47.4 \pm 2.4$ | $55.2 \pm 1.6$ | $47.6 \pm 1.4$ | $54.0 \pm 1.2$ | $49.2 \pm 1.9$ | $\underline{58.4 \pm 0.5}$ | $52.1 \pm 0.6$ | $55.3 \pm 1.7$ | $55.8 \pm 1.1$ | $\mathbf{59.6 \pm 0.6}$ |
| 6 | $58.7 \pm 0.9$ | $56.6 \pm 0.3$ | $54.2 \pm 1.5$ | $60.9 \pm 0.9$ | $54.5 \pm 1.2$ | $60.9 \pm 0.7$ | $54.3 \pm 1.1$ | $\underline{62.2 \pm 0.8}$ | $58.1 \pm 2.0$ | $62.0 \pm 0.7$ | $59.3 \pm 0.6$ | $\mathbf{63.8 \pm 1.3}$ |
| 8 | $62.5 \pm 1.0$ | $61.0 \pm 1.0$ | $60.0 \pm 1.9$ | $64.7 \pm 0.9$ | $58.7 \pm 1.7$ | $64.8 \pm 0.7$ | $58.2 \pm 0.5$ | $65.0 \pm 0.6$ | $59.3 \pm 2.3$ | $\underline{66.1 \pm 1.2}$ | $62.2 \pm 0.7$ | $\mathbf{66.6 \pm 0.5}$ |
| 10 | $64.9 \pm 0.8$ | $64.9 \pm 1.0$ | $64.1 \pm 1.4$ | $67.8 \pm 0.7$ | $63.2 \pm 1.2$ | $67.5 \pm 0.5$ | $61.0 \pm 1.2$ | $66.5 \pm 0.5$ | $61.0 \pm 2.0$ | $\underline{68.0 \pm 0.7}$ | $65.1 \pm 0.6$ | $\mathbf{68.2 \pm 0.9}$ |
| 12 | $66.7 \pm 0.5$ | $67.2 \pm 0.8$ | $66.2 \pm 1.2$ | $\underline{69.9 \pm 1.0}$ | $65.9 \pm 0.8$ | $69.4 \pm 0.6$ | $63.6 \pm 1.2$ | $68.1 \pm 0.6$ | $63.8 \pm 1.9$ | $\mathbf{70.2 \pm 0.9}$ | $66.9 \pm 0.5$ | $69.8 \pm 1.0$ |
| 14 | $68.2 \pm 1.0$ | $69.5 \pm 1.2$ | $68.3 \pm 1.0$ | $\underline{71.6 \pm 0.6}$ | $68.0 \pm 0.8$ | $71.2 \pm 0.5$ | $66.0 \pm 0.9$ | $69.2 \pm 0.8$ | $64.7 \pm 2.2$ | $\mathbf{71.9 \pm 1.1}$ | $68.4 \pm 0.7$ | $71.1 \pm 0.8$ |
| 16 | $70.2 \pm 0.6$ | $70.8 \pm 1.0$ | $70.5 \pm 0.7$ | $\underline{72.8 \pm 0.6}$ | $70.0 \pm 0.5$ | $72.5 \pm 0.9$ | $68.5 \pm 1.2$ | $70.3 \pm 0.7$ | $64.8 \pm 1.9$ | $\mathbf{73.1 \pm 1.2}$ | $69.4 \pm 0.6$ | $72.4 \pm 0.5$ |
| 18 | $71.0 \pm 0.6$ | $72.3 \pm 0.7$ | $71.3 \pm 1.0$ | $\underline{74.1 \pm 0.5}$ | $71.5 \pm 0.5$ | $73.6 \pm 0.5$ | $69.6 \pm 1.2$ | $71.1 \pm 0.6$ | $66.3 \pm 1.6$ | $\mathbf{74.7 \pm 0.8}$ | $70.7 \pm 0.4$ | $73.3 \pm 0.4$ |
| 20 | $72.2 \pm 0.4$ | $73.9 \pm 0.5$ | $73.0 \pm 0.7$ | $\underline{75.5 \pm 0.6}$ | $72.9 \pm 0.5$ | $74.1 \pm 0.6$ | $70.8 \pm 0.7$ | $72.0 \pm 0.4$ | $68.3 \pm 1.1$ | $\mathbf{75.7 \pm 0.7}$ | $72.2 \pm 0.5$ | $74.4 \pm 0.6$ |
| | | | | | | Oxford-IIIT Pets | | | | | | |
| 1 | $47.2 \pm 5.0$ | $47.2 \pm 5.0$ | $47.2 \pm 5.0$ | $47.2 \pm 5.0$ | $47.7 \pm 5.6$ | $47.2 \pm 5.0$ | $47.2 \pm 5.0$ | $69.5 \pm 3.8$ | $\mathbf{70.7 \pm 4.5}$ | $47.2 \pm 5.0$ | $47.2 \pm 5.0$ | $\underline{70.4 \pm 1.6}$ |
| 2 | $67.6 \pm 7.8$ | $60.9 \pm 7.2$ | $59.9 \pm 7.0$ | $70.0 \pm 3.2$ | $61.1 \pm 7.5$ | $67.8 \pm 2.4$ | $63.6 \pm 3.9$ | $\underline{84.6 \pm 1.6}$ | $\mathbf{85.2 \pm 2.7}$ | $68.5 \pm 3.8$ | $77.3 \pm 1.7$ | $84.6 \pm 1.5$ |
| 4 | $80.7 \pm 3.9$ | $78.7 \pm 3.6$ | $77.4 \pm 3.7$ | $84.9 \pm 3.5$ | $77.8 \pm 2.6$ | $84.5 \pm 1.1$ | $77.7 \pm 2.7$ | $89.6 \pm 1.0$ | $\underline{89.8 \pm 1.6}$ | $86.8 \pm 2.5$ | $86.3 \pm 1.9$ | $\mathbf{90.1 \pm 1.1}$ |
| 6 | $87.9 \pm 2.8$ | $88.0 \pm 1.2$ | $86.1 \pm 3.3$ | $89.4 \pm 0.7$ | $86.0 \pm 2.9$ | $89.1 \pm 1.0$ | $84.0 \pm 1.5$ | $91.3 \pm 1.0$ | $92.1 \pm 0.4$ | $90.0 \pm 0.8$ | $89.5 \pm 1.6$ | $\mathbf{92.7 \pm 0.6}$ |
| 8 | $89.5 \pm 2.0$ | $91.1 \pm 0.6$ | $90.1 \pm 1.5$ | $91.4 \pm 0.3$ | $90.1 \pm 1.4$ | $91.5 \pm 0.6$ | $88.4 \pm 1.5$ | $92.6 \pm 0.8$ | $92.6 \pm 0.7$ | $91.6 \pm 0.8$ | $90.0 \pm 0.2$ | $\mathbf{93.4 \pm 0.4}$ |
| 10 | $91.4 \pm 0.6$ | $92.8 \pm 0.6$ | $91.9 \pm 0.5$ | $92.6 \pm 0.6$ | $91.7 \pm 0.4$ | $92.8 \pm 0.3$ | $89.9 \pm 1.4$ | $\underline{93.2 \pm 0.8}$ | $93.0 \pm 1.0$ | $92.7 \pm 0.6$ | $91.2 \pm 0.6$ | $\mathbf{93.7 \pm 0.5}$ |
| 12 | $91.9 \pm 0.6$ | $93.3 \pm 0.5$ | $93.1 \pm 0.7$ | $93.7 \pm 0.7$ | $92.3 \pm 0.8$ | $92.8 \pm 0.6$ | $91.4 \pm 0.8$ | $\underline{93.7 \pm 0.8}$ | $93.0 \pm 0.9$ | $93.0 \pm 0.7$ | $92.0 \pm 0.6$ | $\mathbf{93.9 \pm 0.3}$ |
| 14 | $92.7 \pm 0.4$ | $93.9 \pm 0.4$ | $93.9 \pm 0.5$ | $\underline{94.1 \pm 0.2}$ | $93.2 \pm 0.8$ | $93.6 \pm 0.4$ | $91.6 \pm 0.6$ | $93.9 \pm 0.2$ | $92.7 \pm 0.2$ | $93.7 \pm 0.5$ | $93.1 \pm 0.3$ | $\mathbf{94.2 \pm 0.3}$ |
| 16 | $93.0 \pm 0.6$ | $94.4 \pm 0.2$ | $94.0 \pm 0.2$ | $\mathbf{94.5 \pm 0.2}$ | $93.8 \pm 0.5$ | $94.0 \pm 0.5$ | $92.5 \pm 0.8$ | $94.2 \pm 0.6$ | $92.3 \pm 0.4$ | $94.2 \pm 0.3$ | $93.4 \pm 0.4$ | $\underline{94.5 \pm 0.3}$ |
| 18 | $93.3 \pm 0.7$ | $94.5 \pm 0.3$ | $\mathbf{94.9 \pm 0.2}$ | $94.6 \pm 0.1$ | $94.1 \pm 0.3$ | $94.3 \pm 0.3$ | $93.1 \pm 0.6$ | $94.2 \pm 0.3$ | $92.5 \pm 0.8$ | $94.5 \pm 0.2$ | $93.6 \pm 0.2$ | $\underline{94.6 \pm 0.2}$ |
| 20 | $93.7 \pm 0.4$ | $94.7 \pm 0.1$ | $\underline{94.8 \pm 0.3}$ | $94.8 \pm 0.1$ | $94.7 \pm 0.1$ | $94.5 \pm 0.5$ | $93.2 \pm 0.4$ | $94.1 \pm 0.3$ | $93.1 \pm 0.2$ | $94.8 \pm 0.2$ | $94.0 \pm 0.2$ | $\mathbf{94.8 \pm 0.2}$ |
| | | | | | | Places365 | | | | | | |
| 1 | $18.1 \pm 0.5$ | $18.1 \pm 0.5$ | $18.1 \pm 0.5$ | $18.1 \pm 0.5$ | $18.1 \pm 0.5$ | $18.1 \pm 0.5$ | $18.1 \pm 0.5$ | $\underline{26.7 \pm 0.7}$ | - | $18.1 \pm 0.5$ | $18.1 \pm 0.5$ | $\mathbf{30.7 \pm 0.6}$ |
| 2 | $24.9 \pm 0.9$ | $21.7 \pm 0.4$ | $20.7 \pm 0.3$ | $25.3 \pm 0.5$ | $22.8 \pm 0.5$ | $25.9 \pm 0.5$ | $20.4 \pm 0.7$ | $\underline{32.4 \pm 0.3}$ | - | $25.1 \pm 0.6$ | $32.1 \pm 0.5$ | $\mathbf{35.3 \pm 0.7}$ |
| 4 | $32.4 \pm 0.7$ | $27.9 \pm 0.5$ | $26.5 \pm 0.2$ | $33.0 \pm 0.4$ | $29.0 \pm 0.8$ | $33.3 \pm 0.4$ | $24.0 \pm 0.6$ | $\underline{37.2 \pm 0.4}$ | - | $32.9 \pm 0.4$ | $36.6 \pm 0.3$ | $\mathbf{38.8 \pm 0.4}$ |
| 6 | $36.5 \pm 0.6$ | $31.9 \pm 0.8$ | $30.5 \pm 0.3$ | $36.9 \pm 0.4$ | $32.7 \pm 0.6$ | $37.1 \pm 0.3$ | $27.3 \pm 0.4$ | $\underline{38.7 \pm 0.3}$ | - | $36.8 \pm 0.4$ | $38.5 \pm 0.4$ | $\mathbf{40.6 \pm 0.3}$ |
| 8 | $38.9 \pm 0.5$ | $34.9 \pm 0.7$ | $33.3 \pm 0.3$ | $39.5 \pm 0.3$ | $35.3 \pm 0.6$ | $\underline{39.6 \pm 0.1}$ | $29.9 \pm 0.6$ | $39.6 \pm 0.3$ | - | $39.3 \pm 0.6$ | $39.4 \pm 0.4$ | $\mathbf{41.6 \pm 0.5}$ |
| 10 | $40.9 \pm 0.3$ | $36.9 \pm 0.4$ | $35.3 \pm 0.2$ | $41.1 \pm 0.4$ | $37.0 \pm 0.3$ | $41.3 \pm 0.2$ | $31.9 \pm 0.6$ | $40.3 \pm 0.5$ | - | $\underline{41.4 \pm 0.4}$ | $40.1 \pm 0.2$ | $\mathbf{42.5 \pm 0.3}$ |
| 12 | $41.5 \pm 0.4$ | $37.6 \pm 0.2$ | $36.1 \pm 0.3$ | $41.7 \pm 0.4$ | $37.4 \pm 0.3$ | $41.8 \pm 0.3$ | $32.4 \pm 0.7$ | $40.4 \pm 0.4$ | - | $\underline{41.9 \pm 0.2}$ | $40.0 \pm 0.2$ | $\mathbf{42.3 \pm 0.3}$ |
| 14 | $42.8 \pm 0.4$ | $39.0 \pm 0.4$ | $37.5 \pm 0.3$ | $43.2 \pm 0.4$ | $39.1 \pm 0.3$ | $\mathbf{43.4 \pm 0.2}$ | $34.2 \pm 0.4$ | $40.9 \pm 0.7$ | - | $43.3 \pm 0.2$ | $40.9 \pm 0.2$ | $\underline{43.3 \pm 0.2}$ |
| 16 | $43.8 \pm 0.4$ | $40.2 \pm 0.5$ | $38.6 \pm 0.2$ | $44.1 \pm 0.2$ | $40.2 \pm 0.2$ | $\underline{44.1 \pm 0.3}$ | $35.3 \pm 0.4$ | $41.6 \pm 0.1$ | - | $\mathbf{44.3 \pm 0.1}$ | $41.3 \pm 0.2$ | $44.0 \pm 0.1$ |
| 18 | $44.4 \pm 0.4$ | $41.1 \pm 0.2$ | $39.6 \pm 0.3$ | $44.8 \pm 0.4$ | $41.0 \pm 0.3$ | $\underline{44.9 \pm 0.3}$ | $36.6 \pm 0.5$ | $42.0 \pm 0.5$ | - | $\mathbf{45.1 \pm 0.1}$ | $42.0 \pm 0.3$ | $44.4 \pm 0.1$ |
| 20 | $45.0 \pm 0.3$ | $41.6 \pm 0.4$ | $40.4 \pm 0.3$ | $\underline{45.5 \pm 0.3}$ | $41.7 \pm 0.3$ | $45.5 \pm 0.3$ | $37.7 \pm 0.5$ | $42.4 \pm 0.3$ | - | $\mathbf{45.6 \pm 0.2}$ | $42.3 \pm 0.2$ | $44.8 \pm 0.2$ |

Table 6: Test set accuracy with standard deviations (average of 5 random seeds) at certain iterations $t$ during AL on out-of-domain biomedical datasets including Blood Smear Acevedo et al. (2020), Diabetic Retinopathy Kaggle & EyePacs (2015), IICBU HeLa Murphy et al. (2000), and Skin cancer Tschandl et al. (2018) (from top to bottom) with DINOv2 ViT-g14 as the feature extractor $f$. Probcover was intractable on Patch Camelyon with the resources available, and thus not performed (-). Cells are color coded according to the magnitude of improvement in mean accuracy over the Random baseline.

| $t$ | Random | Uncertainty | Entropy | Margins | BALD | pBALD | Coreset | Typiclust | ProbCover | BADGE | Alfamix | DropQuery (Ours) |
|---|---|---|---|---|---|---|---|---|---|---|---|---|
| | | | | | | Blood Smear | | | | | | |
| 1 | 40.8 ± 3.0 | 40.8 ± 3.0 | 40.8 ± 3.0 | 40.8 ± 3.0 | 40.8 ± 3.0 | 40.8 ± 3.0 | 40.8 ± 3.0 | **48.7 ± 9.8** | 44.2 ± 17.9 | 40.8 ± 3.0 | 40.8 ± 3.0 | 39.9 ± 13.3 |
| 2 | 56.6 ± 2.5 | 50.4 ± 3.0 | 52.7 ± 2.8 | 54.6 ± 4.0 | 54.6 ± 6.0 | 55.1 ± 5.6 | 33.2 ± 5.6 | 63.5 ± 4.0 | 62.4 ± 1.1 | 55.7 ± 4.1 | **64.0 ± 2.6** | 63.3 ± 2.5 |
| 4 | 69.8 ± 4.3 | 65.8 ± 5.8 | 64.4 ± 5.3 | 69.8 ± 5.0 | 69.4 ± 4.9 | 69.6 ± 2.1 | 36.7 ± 7.4 | 73.4 ± 2.7 | 72.5 ± 0.8 | 69.5 ± 1.8 | 75.2 ± 3.0 | **77.2 ± 2.4** |
| 6 | 75.6 ± 4.4 | 73.1 ± 3.5 | 74.8 ± 2.1 | 77.1 ± 5.0 | 73.2 ± 3.5 | 75.5 ± 2.7 | 46.8 ± 6.8 | 78.9 ± 2.1 | 80.3 ± 0.4 | 76.4 ± 2.2 | **81.6 ± 2.3** | 80.3 ± 2.0 |
| 8 | 80.1 ± 2.6 | 78.8 ± 3.9 | 77.2 ± 2.0 | 80.9 ± 2.4 | 77.5 ± 2.7 | 77.7 ± 2.9 | 53.9 ± 7.0 | 80.8 ± 2.1 | 82.7 ± 0.6 | 81.2 ± 0.9 | **84.7 ± 1.4** | 83.4 ± 0.8 |
| 10 | 80.7 ± 1.4 | 71.8 ± 14.5 | 79.3 ± 3.3 | 84.7 ± 1.6 | 80.3 ± 3.7 | 80.3 ± 2.8 | 60.8 ± 6.3 | 83.7 ± 0.6 | 83.4 ± 0.7 | 84.2 ± 2.0 | **87.0 ± 0.5** | 85.0 ± 0.9 |
| 12 | 82.0 ± 2.3 | 83.0 ± 2.5 | 79.4 ± 2.1 | 87.6 ± 0.4 | 82.8 ± 3.8 | 83.6 ± 1.7 | 65.7 ± 6.1 | 84.7 ± 0.9 | 84.4 ± 0.5 | 86.3 ± 1.0 | **88.0 ± 1.2** | 85.7 ± 0.9 |
| 14 | 82.8 ± 2.3 | 83.8 ± 1.5 | 82.1 ± 1.5 | 88.0 ± 0.4 | 85.2 ± 2.0 | 84.9 ± 2.0 | 68.2 ± 2.4 | 84.7 ± 1.5 | 85.6 ± 0.5 | 87.7 ± 0.8 | **89.3 ± 1.1** | 85.6 ± 1.3 |
| 16 | 83.7 ± 2.2 | 84.2 ± 4.6 | 81.9 ± 2.7 | 89.0 ± 1.0 | 86.2 ± 1.9 | 85.7 ± 2.1 | 70.5 ± 1.6 | 86.2 ± 0.6 | 85.7 ± 0.7 | 88.4 ± 1.1 | **89.7 ± 1.8** | 87.2 ± 0.7 |
| 18 | 85.6 ± 1.1 | 86.4 ± 2.5 | 84.1 ± 2.2 | 89.9 ± 1.1 | 85.3 ± 4.3 | 86.7 ± 1.3 | 73.3 ± 4.4 | 87.1 ± 0.3 | 85.2 ± 0.4 | 89.2 ± 0.9 | **90.5 ± 1.0** | 87.3 ± 0.8 |
| 20 | 85.4 ± 3.1 | 87.6 ± 1.5 | 84.9 ± 3.7 | 91.2 ± 0.3 | 88.1 ± 1.4 | 87.8 ± 1.7 | 74.4 ± 2.3 | 88.3 ± 0.4 | 86.2 ± 0.3 | 89.8 ± 0.9 | **91.2 ± 0.8** | 87.8 ± 0.5 |
| | | | | | | Diabetic Retinopathy | | | | | | |
| 1 | 43.3 ± 11.6 | 43.3 ± 11.6 | 43.3 ± 11.6 | 43.3 ± 11.6 | 43.3 ± 11.6 | 43.3 ± 11.6 | 43.3 ± 11.6 | 46.9 ± 4.0 | 56.3 ± 1.1 | 43.3 ± 11.6 | 43.3 ± 11.6 | **57.2 ± 0.9** |
| 2 | 51.0 ± 4.4 | 43.6 ± 6.1 | 44.9 ± 8.9 | 47.9 ± 3.1 | 40.9 ± 9.5 | 45.6 ± 10.0 | 42.2 ± 8.5 | 53.4 ± 3.3 | 53.2 ± 1.2 | 48.9 ± 6.0 | 52.2 ± 3.6 | **57.7 ± 4.0** |
| 4 | 54.2 ± 3.6 | 52.6 ± 6.8 | 52.3 ± 6.8 | 55.5 ± 2.5 | 53.0 ± 4.5 | 58.1 ± 3.1 | 45.5 ± 7.3 | 59.7 ± 4.9 | 57.0 ± 1.0 | 57.0 ± 4.4 | 58.8 ± 3.1 | **61.2 ± 2.1** |
| 6 | 58.1 ± 2.2 | 56.4 ± 4.6 | 57.1 ± 4.7 | 59.0 ± 3.0 | 57.2 ± 5.6 | 57.8 ± 1.6 | 48.8 ± 4.8 | 62.4 ± 3.4 | 62.1 ± 1.8 | 59.7 ± 2.6 | **62.9 ± 2.2** | 62.8 ± 3.1 |
| 8 | 59.7 ± 2.5 | 58.0 ± 1.9 | 57.6 ± 5.7 | 60.4 ± 2.6 | 58.7 ± 2.8 | 60.9 ± 2.4 | 46.9 ± 3.7 | 62.5 ± 2.1 | 60.4 ± 0.6 | 62.6 ± 1.7 | 63.5 ± 3.0 | **64.9 ± 0.7** |
| 10 | 63.1 ± 1.0 | 58.6 ± 3.0 | 57.5 ± 6.7 | 62.5 ± 1.8 | 60.4 ± 5.5 | 61.9 ± 1.8 | 45.1 ± 6.3 | 63.5 ± 1.7 | 63.1 ± 0.6 | 62.5 ± 2.8 | **64.6 ± 2.8** | 64.4 ± 1.0 |
| 12 | 64.0 ± 2.2 | 59.2 ± 1.7 | 59.1 ± 8.2 | 62.9 ± 3.5 | 59.8 ± 4.2 | 63.7 ± 1.6 | 51.7 ± 1.7 | **65.7 ± 1.4** | 61.8 ± 0.7 | 63.2 ± 3.6 | 65.3 ± 3.0 | 65.3 ± 1.9 |
| 14 | 64.7 ± 1.5 | 59.0 ± 4.8 | 58.8 ± 5.8 | 64.5 ± 2.7 | 63.0 ± 3.4 | 63.4 ± 2.7 | 51.8 ± 3.6 | **65.7 ± 2.5** | 62.8 ± 1.4 | 63.4 ± 2.0 | 64.3 ± 2.0 | 65.2 ± 1.7 |
| 16 | 65.8 ± 1.6 | 58.6 ± 4.7 | 61.2 ± 5.7 | 62.8 ± 4.4 | 63.3 ± 3.3 | 64.8 ± 1.9 | 53.0 ± 3.4 | 65.9 ± 0.9 | 64.3 ± 1.2 | 63.9 ± 1.9 | 65.0 ± 2.2 | **66.1 ± 1.6** |
| 18 | 65.4 ± 1.5 | 60.3 ± 4.2 | 61.1 ± 5.5 | 64.4 ± 3.4 | 63.8 ± 2.2 | 64.4 ± 1.3 | 52.7 ± 4.1 | 66.5 ± 1.2 | 63.8 ± 0.3 | 64.2 ± 3.0 | 66.0 ± 1.7 | **66.7 ± 1.2** |
| 20 | 65.8 ± 0.7 | 61.1 ± 3.0 | 61.7 ± 3.3 | 65.5 ± 3.1 | 65.4 ± 1.9 | 65.3 ± 1.3 | 54.2 ± 3.1 | 65.0 ± 2.2 | 62.7 ± 0.5 | 65.8 ± 3.2 | **66.4 ± 2.2** | 66.2 ± 0.9 |
| | | | | | | IICBU Hela | | | | | | |
| 1 | 26.5 ± 4.6 | 26.5 ± 4.6 | 26.5 ± 4.6 | 26.5 ± 4.6 | 26.5 ± 4.6 | 26.5 ± 4.6 | 26.5 ± 4.6 | **40.5 ± 1.8** | 19.2 ± 2.8 | 26.5 ± 4.6 | 26.5 ± 4.6 | 37.0 ± 3.9 |
| 2 | 42.1 ± 4.1 | 34.0 ± 3.0 | 31.3 ± 6.5 | 40.2 ± 6.7 | 34.8 ± 8.2 | 36.4 ± 5.7 | 39.8 ± 4.0 | 43.7 ± 3.1 | 30.1 ± 7.3 | 41.4 ± 4.2 | 45.0 ± 2.4 | **46.7 ± 4.8** |
| 4 | 51.3 ± 3.5 | 44.2 ± 9.8 | 38.3 ± 2.9 | 53.2 ± 3.9 | 49.2 ± 6.1 | 49.8 ± 1.8 | 42.4 ± 3.5 | 52.5 ± 2.9 | 49.9 ± 2.2 | 49.4 ± 2.4 | 53.4 ± 6.6 | **54.8 ± 3.1** |
| 6 | 56.5 ± 2.7 | 52.4 ± 6.8 | 48.9 ± 5.8 | 59.9 ± 3.4 | 54.9 ± 3.9 | 60.1 ± 2.5 | 45.4 ± 3.8 | 59.3 ± 3.7 | 49.7 ± 1.4 | 57.8 ± 3.9 | 57.3 ± 4.5 | **61.5 ± 2.1** |
| 8 | 60.5 ± 5.9 | 58.8 ± 8.9 | 55.3 ± 9.3 | **64.5 ± 1.6** | 57.6 ± 1.8 | 64.4 ± 3.5 | 48.2 ± 2.3 | 63.6 ± 4.3 | 56.3 ± 1.6 | 59.5 ± 3.4 | 63.9 ± 5.6 | 64.3 ± 4.1 |
| 10 | 61.3 ± 3.0 | 61.7 ± 10.7 | 60.1 ± 8.4 | **68.8 ± 1.8** | 63.9 ± 1.5 | 67.3 ± 3.1 | 52.0 ± 0.9 | 63.1 ± 3.2 | 59.3 ± 3.0 | 64.4 ± 2.1 | 66.2 ± 4.9 | 67.3 ± 2.2 |
| 12 | 64.9 ± 4.3 | 65.0 ± 9.7 | 61.3 ± 5.6 | 70.1 ± 4.3 | 64.3 ± 1.4 | **72.0 ± 2.8** | 54.7 ± 3.0 | 66.5 ± 4.9 | 57.7 ± 2.1 | 68.1 ± 2.4 | 69.2 ± 5.0 | 69.5 ± 2.7 |
| 14 | 66.5 ± 4.1 | 66.4 ± 7.2 | 63.8 ± 6.3 | 71.7 ± 3.0 | 66.8 ± 4.5 | **72.8 ± 3.1** | 55.6 ± 2.8 | 67.7 ± 5.2 | 62.1 ± 2.6 | 69.5 ± 2.2 | 71.6 ± 5.9 | 71.6 ± 1.7 |
| 16 | 68.4 ± 3.2 | 70.2 ± 5.0 | 65.3 ± 7.3 | **74.3 ± 1.3** | 69.4 ± 4.8 | 73.5 ± 2.8 | 59.5 ± 2.0 | 68.7 ± 4.5 | 63.9 ± 3.0 | 72.9 ± 2.1 | 72.4 ± 4.0 | 72.1 ± 1.3 |
| 18 | 68.6 ± 2.2 | 68.0 ± 5.7 | 63.5 ± 4.3 | 74.9 ± 1.7 | 69.9 ± 1.0 | **76.3 ± 3.1** | 60.2 ± 2.8 | 70.5 ± 1.1 | 64.0 ± 1.7 | 73.8 ± 1.5 | 74.8 ± 3.5 | 74.0 ± 2.3 |
| 20 | 71.3 ± 2.5 | 71.4 ± 5.0 | 65.5 ± 5.7 | 74.7 ± 1.3 | 73.5 ± 4.0 | **77.2 ± 1.9** | 62.5 ± 2.1 | 69.4 ± 2.0 | 68.3 ± 1.4 | 74.0 ± 3.1 | 76.6 ± 3.5 | 73.3 ± 2.0 |
| | | | | | | Skin cancer | | | | | | |
| 1 | 48.6 ± 3.4 | 48.6 ± 3.4 | 48.6 ± 3.4 | 48.6 ± 3.4 | 48.6 ± 3.4 | 48.6 ± 3.4 | 48.6 ± 3.4 | 33.7 ± 3.7 | 17.3 ± 11.3 | 48.6 ± 3.4 | 48.6 ± 3.4 | 45.5 ± 15.6 |
| 2 | 51.4 ± 6.8 | 53.0 ± 4.8 | 50.1 ± 7.9 | 52.2 ± 3.3 | 53.3 ± 9.8 | 51.8 ± 7.6 | 51.9 ± 11.8 | 48.7 ± 2.0 | 14.2 ± 15.8 | **56.2 ± 4.4** | 52.6 ± 1.5 | 54.1 ± 7.0 |
| 4 | 49.8 ± 8.0 | 55.2 ± 6.3 | 52.9 ± 6.5 | 57.4 ± 5.0 | 57.6 ± 6.9 | 58.8 ± 3.5 | 56.3 ± 4.4 | 54.2 ± 3.3 | 29.7 ± 17.3 | **59.2 ± 3.6** | 55.4 ± 3.4 | 58.3 ± 5.0 |
| 6 | 53.9 ± 5.7 | 57.9 ± 5.7 | 54.4 ± 6.4 | 59.6 ± 5.2 | 61.9 ± 3.8 | 60.5 ± 2.4 | 61.3 ± 4.0 | 52.9 ± 4.4 | **63.8 ± 2.0** | 61.5 ± 3.6 | 54.6 ± 5.2 | 61.3 ± 5.5 |
| 8 | 55.5 ± 3.6 | 59.7 ± 4.1 | 56.3 ± 7.5 | 62.7 ± 3.5 | **63.9 ± 3.4** | 62.2 ± 3.7 | 61.5 ± 6.2 | 56.4 ± 2.3 | 51.9 ± 13.0 | 62.6 ± 3.8 | 56.5 ± 5.9 | 63.6 ± 2.3 |
| 10 | 58.6 ± 1.8 | 61.2 ± 3.6 | 59.0 ± 5.7 | 63.3 ± 2.4 | **65.4 ± 2.6** | 63.8 ± 2.6 | 62.0 ± 4.5 | 57.0 ± 3.0 | 59.3 ± 9.1 | 64.1 ± 3.6 | 58.9 ± 6.0 | 64.4 ± 2.4 |
| 12 | 58.9 ± 1.3 | 62.4 ± 3.5 | 60.3 ± 5.8 | 63.9 ± 2.5 | **66.9 ± 2.3** | 65.8 ± 1.9 | 65.0 ± 2.3 | 58.0 ± 2.4 | 60.2 ± 11.8 | 66.1 ± 2.6 | 60.0 ± 5.5 | 66.0 ± 1.7 |
| 14 | 60.0 ± 2.1 | 63.9 ± 3.5 | 61.2 ± 7.2 | 65.7 ± 2.9 | **68.4 ± 1.7** | 67.1 ± 0.6 | 65.1 ± 2.8 | 58.5 ± 3.2 | 63.4 ± 5.2 | 67.1 ± 2.0 | 61.8 ± 5.0 | 66.7 ± 1.9 |
| 16 | 60.4 ± 1.8 | 65.0 ± 2.8 | 64.8 ± 4.5 | 67.6 ± 2.4 | 67.0 ± 2.6 | 67.1 ± 1.4 | 65.5 ± 3.3 | 59.4 ± 2.2 | 59.1 ± 12.7 | 67.5 ± 2.4 | 64.6 ± 3.5 | **68.3 ± 1.6** |
| 18 | 60.9 ± 1.5 | 65.1 ± 3.9 | 62.9 ± 6.2 | 67.7 ± 1.7 | 67.1 ± 4.2 | 67.9 ± 1.3 | 64.6 ± 3.4 | 60.6 ± 3.1 | 62.7 ± 9.6 | 68.6 ± 2.5 | 66.7 ± 2.8 | **68.6 ± 2.2** |
| 20 | 62.0 ± 1.9 | 66.0 ± 2.6 | 67.3 ± 2.7 | 68.9 ± 1.5 | 69.4 ± 2.2 | 68.7 ± 0.7 | 64.8 ± 3.6 | 60.6 ± 3.2 | 65.8 ± 4.2 | **69.5 ± 1.9** | 67.7 ± 2.8 | 68.8 ± 1.4 |
| | | | | | | Colorectal Histology | | | | | | |
| 1 | 42.7 ± 4.1 | 42.7 ± 4.1 | 42.7 ± 4.1 | 42.7 ± 4.1 | 42.7 ± 4.1 | 42.7 ± 4.1 | 42.7 ± 4.1 | 47.5 ± 5.0 | 32.7 ± 4.0 | 42.7 ± 4.1 | 42.7 ± 4.1 | **47.9 ± 5.7** |
| 2 | 53.4 ± 6.3 | 48.1 ± 3.5 | 45.6 ± 2.0 | 55.4 ± 6.0 | 46.9 ± 5.5 | 53.1 ± 6.3 | 47.1 ± 6.5 | 54.2 ± 4.2 | 39.2 ± 1.2 | 57.1 ± 4.3 | **57.4 ± 4.6** | 55.7 ± 8.9 |
| 4 | 65.4 ± 3.0 | 58.5 ± 5.4 | 55.4 ± 5.1 | 67.5 ± 6.9 | 56.1 ± 7.2 | 69.3 ± 5.0 | 49.5 ± 4.1 | 66.9 ± 1.6 | 44.8 ± 3.7 | 72.7 ± 2.4 | 68.9 ± 5.3 | **74.2 ± 4.9** |
| 6 | 68.4 ± 5.9 | 70.1 ± 8.0 | 63.0 ± 5.7 | 74.2 ± 6.6 | 64.1 ± 7.4 | 72.7 ± 3.0 | 51.2 ± 9.1 | 75.4 ± 1.3 | 49.4 ± 5.2 | 76.3 ± 2.6 | 72.5 ± 7.8 | **77.3 ± 1.8** |
| 8 | 74.7 ± 5.1 | 74.8 ± 9.1 | 67.5 ± 5.6 | 78.7 ± 5.4 | 71.5 ± 6.9 | 78.1 ± 1.5 | 52.3 ± 4.9 | 76.7 ± 0.8 | 49.7 ± 5.2 | 79.7 ± 1.5 | 76.0 ± 7.0 | **80.7 ± 1.4** |
| 10 | 76.7 ± 3.8 | 76.9 ± 8.0 | 72.4 ± 8.3 | 81.2 ± 2.0 | 78.5 ± 2.8 | 79.4 ± 2.0 | 53.4 ± 6.0 | 76.9 ± 3.6 | 51.5 ± 4.3 | 83.3 ± 1.4 | 80.6 ± 2.9 | 81.8 ± 1.3 |
| 12 | 79.1 ± 2.5 | 76.9 ± 8.5 | 75.1 ± 6.6 | 82.1 ± 1.5 | 80.2 ± 0.7 | 81.2 ± 1.3 | 56.2 ± 4.4 | 80.6 ± 1.6 | 54.3 ± 2.4 | 83.3 ± 1.7 | 82.5 ± 3.4 | 82.8 ± 1.1 |
| 14 | 77.6 ± 5.2 | 77.0 ± 7.6 | 76.8 ± 7.1 | 84.1 ± 1.2 | 81.2 ± 1.8 | 83.3 ± 1.9 | 55.3 ± 5.1 | 81.9 ± 1.3 | 55.5 ± 4.8 | 84.3 ± 1.4 | 84.2 ± 2.8 | **84.4 ± 1.0** |
| 16 | 80.5 ± 2.7 | 77.8 ± 7.0 | 78.2 ± 5.3 | **85.4 ± 1.7** | 82.4 ± 2.0 | 84.4 ± 1.1 | 54.2 ± 5.8 | 81.9 ± 1.6 | 56.0 ± 4.1 | 84.9 ± 1.3 | 85.4 ± 1.6 | 85.0 ± 0.7 |
| 18 | 81.4 ± 2.1 | 79.1 ± 6.5 | 80.9 ± 5.8 | 86.0 ± 1.8 | 84.2 ± 1.7 | 85.2 ± 1.5 | 55.0 ± 6.9 | 83.6 ± 0.9 | 65.9 ± 11.3 | 85.8 ± 0.8 | **86.3 ± 1.6** | 85.7 ± 0.9 |
| 20 | 82.3 ± 3.0 | 80.9 ± 7.0 | 82.7 ± 2.9 | **87.0 ± 0.6** | 85.4 ± 1.7 | 85.7 ± 1.4 | 58.3 ± 6.7 | 84.2 ± 0.8 | 80.9 ± 2.1 | 85.2 ± 1.7 | 86.8 ± 1.1 | 85.9 ± 0.8 |
| | | | | | | Patch Camelyon | | | | | | |
| 1 | 52.0 ± 7.7 | 52.0 ± 7.7 | 52.0 ± 7.7 | 52.0 ± 7.7 | 52.0 ± 7.7 | 52.0 ± 7.7 | 52.0 ± 7.7 | 47.4 ± 6.0 | - | 52.0 ± 7.7 | 52.0 ± 7.7 | 47.8 ± 5.0 |
| 2 | 55.3 ± 8.4 | **60.5 ± 7.4** | 54.0 ± 11.5 | 56.4 ± 8.2 | 60.2 ± 3.0 | 55.2 ± 11.1 | 55.3 ± 7.9 | 56.2 ± 0.5 | - | 49.5 ± 12.9 | 58.9 ± 3.9 | 58.3 ± 9.1 |
| 4 | 63.4 ± 7.3 | 59.6 ± 13.1 | 62.4 ± 4.4 | 60.6 ± 11.9 | 54.4 ± 6.3 | **66.2 ± 2.8** | 52.3 ± 2.5 | 59.5 ± 1.0 | - | 64.1 ± 4.7 | 66.0 ± 2.6 | 64.9 ± 1.7 |
| 6 | 62.7 ± 6.5 | 60.7 ± 10.3 | 65.9 ± 2.7 | 61.4 ± 9.4 | 56.0 ± 7.3 | **68.9 ± 2.2** | 52.1 ± 3.3 | 59.8 ± 0.5 | - | 60.0 ± 11.3 | 64.6 ± 1.9 | 65.1 ± 6.3 |
| 8 | 70.3 ± 3.9 | 66.6 ± 5.2 | 65.9 ± 4.0 | 65.6 ± 4.2 | 57.5 ± 3.0 | 69.9 ± 3.1 | 53.2 ± 2.6 | 60.1 ± 1.0 | - | 64.8 ± 6.9 | 61.3 ± 6.9 | 65.8 ± 3.7 |
| 10 | 68.9 ± 4.2 | 67.5 ± 3.7 | 68.6 ± 4.7 | 64.2 ± 5.4 | 60.3 ± 2.8 | **71.7 ± 1.5** | 50.8 ± 2.6 | 60.3 ± 0.8 | - | 62.9 ± 10.8 | 65.6 ± 9.7 | 64.1 ± 5.1 |
| 12 | 71.5 ± 4.1 | 69.2 ± 4.9 | 69.1 ± 4.1 | 65.6 ± 2.8 | 61.1 ± 4.0 | 69.5 ± 4.9 | 52.3 ± 5.4 | 60.8 ± 0.9 | - | 68.4 ± 9.9 | 67.8 ± 5.4 | 64.1 ± 3.9 |
| 14 | 74.8 ± 2.7 | 68.9 ± 5.3 | 70.4 ± 4.4 | 67.9 ± 4.8 | 61.8 ± 5.2 | 71.2 ± 4.8 | 51.6 ± 4.4 | 60.9 ± 0.8 | - | 66.3 ± 9.0 | 65.6 ± 11.2 | 60.6 ± 10.5 |
| 16 | 73.5 ± 3.8 | 69.9 ± 5.3 | 69.2 ± 4.6 | 68.5 ± 3.8 | 59.9 ± 2.7 | 73.2 ± 3.5 | 50.4 ± 3.4 | 60.8 ± 0.8 | - | 70.6 ± 6.0 | 63.5 ± 9.3 | 65.4 ± 6.3 |
| 18 | 72.4 ± 4.7 | 70.2 ± 7.5 | 71.9 ± 5.7 | 67.4 ± 5.6 | 63.5 ± 6.6 | **74.0 ± 1.9** | 52.5 ± 2.6 | 60.6 ± 0.6 | - | 70.8 ± 5.5 | 70.1 ± 1.6 | 58.6 ± 8.5 |
| 20 | 73.6 ± 3.2 | 70.6 ± 7.1 | 69.1 ± 5.0 | 68.6 ± 5.1 | 60.5 ± 5.2 | **74.6 ± 3.4** | 50.4 ± 2.2 | 60.8 ± 0.9 | - | 73.8 ± 4.0 | 68.2 ± 4.4 | 66.0 ± 7.1 |

Table 7: Test set accuracy with standard deviations (average of 5 random seeds) at certain iterations $t$ during AL on CIFAR100 Krizhevsky (2009), Food101 Bossard et al. (2014), ImageNet-100 Gansbeke et al. (2020), and DomainNet-Real Peng et al. (2019) (from top to bottom) with an ImageNet-1K pretrained Masked AutoEncoder ViT-H/14 He et al. (2022) as the feature extractor $f$. Typiclust, and our proposed **DropQuery** use their own respective initialization strategies for the cold-start. Cells are color coded according to the magnitude of improvement in mean accuracy over the Random baseline. Green cells are positive with better performance than random whereas red cells are negative with worse performance than random. **Bold** values represent the **first place** mean accuracy at iteration $t$ with the second place value underlined.

| $t$ | Random | Margins | pBALD | BADGE | Typiclust | Alfamix | DropQuery (Ours) |
|---|---|---|---|---|---|---|---|
| | | | | CIFAR100 | | | |
| 1 | $14.2 \pm 1.3$ | $14.2 \pm 1.3$ | $14.2 \pm 1.3$ | $14.2 \pm 1.3$ | $\mathbf{16.2 \pm 0.6}$ | $14.2 \pm 1.3$ | $13.0 \pm 0.6$ |
| 2 | $21.0 \pm 1.5$ | $20.2 \pm 1.3$ | $21.4 \pm 1.1$ | $21.8 \pm 0.6$ | $21.7 \pm 0.6$ | $\mathbf{23.7 \pm 0.9}$ | $\underline{22.9 \pm 1.5}$ |
| 4 | $31.0 \pm 1.3$ | $30.4 \pm 0.8$ | $30.6 \pm 0.9$ | $31.3 \pm 1.5$ | $29.2 \pm 0.8$ | $\underline{31.9 \pm 0.9}$ | $\mathbf{33.1 \pm 0.9}$ |
| 6 | $37.5 \pm 1.0$ | $37.6 \pm 0.9$ | $\underline{38.5 \pm 1.2}$ | $\mathbf{38.6 \pm 1.0}$ | $35.0 \pm 0.8$ | $37.4 \pm 0.9$ | $38.4 \pm 0.8$ |
| 8 | $42.1 \pm 1.2$ | $42.6 \pm 0.9$ | $\underline{42.8 \pm 1.0}$ | $\mathbf{43.4 \pm 0.9}$ | $39.1 \pm 0.7$ | $40.8 \pm 1.4$ | $42.5 \pm 0.7$ |
| 10 | $45.8 \pm 0.8$ | $45.9 \pm 1.0$ | $\underline{46.3 \pm 0.5}$ | $\mathbf{46.9 \pm 0.9}$ | $41.3 \pm 0.9$ | $44.4 \pm 2.0$ | $44.7 \pm 0.3$ |
| 12 | $48.7 \pm 0.8$ | $48.1 \pm 0.7$ | $\underline{49.2 \pm 0.8}$ | $\mathbf{49.7 \pm 0.6}$ | $43.8 \pm 0.8$ | $46.9 \pm 1.0$ | $47.0 \pm 0.3$ |
| 14 | $50.9 \pm 0.6$ | $50.4 \pm 0.9$ | $\underline{51.6 \pm 0.6}$ | $\mathbf{51.9 \pm 0.3}$ | $44.8 \pm 0.7$ | $49.1 \pm 1.3$ | $48.6 \pm 0.3$ |
| 16 | $51.9 \pm 0.9$ | $52.1 \pm 0.4$ | $\underline{53.3 \pm 1.3}$ | $\mathbf{53.5 \pm 0.6}$ | $46.7 \pm 0.4$ | $50.9 \pm 1.5$ | $50.3 \pm 0.4$ |
| 18 | $53.2 \pm 0.3$ | $54.1 \pm 0.3$ | $\underline{54.3 \pm 1.0}$ | $\mathbf{55.3 \pm 0.4}$ | $47.8 \pm 0.6$ | $52.2 \pm 0.8$ | $51.4 \pm 0.8$ |
| 20 | $55.1 \pm 0.4$ | $55.2 \pm 0.4$ | $\underline{55.9 \pm 0.4}$ | $\mathbf{56.1 \pm 0.5}$ | $49.3 \pm 0.6$ | $53.9 \pm 0.8$ | $52.0 \pm 0.9$ |
| | | | | Food101 | | | |
| 1 | $\mathbf{6.6 \pm 0.8}$ | $\mathbf{6.6 \pm 0.8}$ | $\mathbf{6.6 \pm 0.8}$ | $\mathbf{6.6 \pm 0.8}$ | $\underline{6.4 \pm 0.4}$ | $\mathbf{6.6 \pm 0.8}$ | $6.1 \pm 0.4$ |
| 2 | $10.3 \pm 1.0$ | $10.6 \pm 1.1$ | $11.0 \pm 0.6$ | $10.5 \pm 0.6$ | $9.8 \pm 0.3$ | $\mathbf{12.4 \pm 0.8}$ | $\underline{12.0 \pm 0.7}$ |
| 4 | $17.6 \pm 0.6$ | $17.8 \pm 0.8$ | $17.9 \pm 0.6$ | $17.9 \pm 0.8$ | $13.8 \pm 0.5$ | $\underline{18.5 \pm 0.6}$ | $\mathbf{18.5 \pm 0.3}$ |
| 6 | $23.2 \pm 0.4$ | $23.0 \pm 0.8$ | $\mathbf{23.6 \pm 0.7}$ | $\underline{23.5 \pm 1.1}$ | $20.1 \pm 0.3$ | $23.4 \pm 0.6$ | $23.0 \pm 0.2$ |
| 8 | $27.5 \pm 1.2$ | $27.2 \pm 0.8$ | $\mathbf{27.6 \pm 0.4}$ | $\underline{27.6 \pm 1.0}$ | $24.0 \pm 0.5$ | $26.8 \pm 0.7$ | $26.6 \pm 0.5$ |
| 10 | $\underline{31.0 \pm 0.6}$ | $30.4 \pm 0.4$ | $30.9 \pm 1.3$ | $\mathbf{31.1 \pm 1.4}$ | $27.0 \pm 1.1$ | $29.3 \pm 0.9$ | $29.4 \pm 0.5$ |
| 12 | $33.7 \pm 0.7$ | $33.1 \pm 0.7$ | $\mathbf{34.1 \pm 1.0}$ | $\underline{34.0 \pm 0.7}$ | $28.6 \pm 1.0$ | $32.1 \pm 0.7$ | $31.8 \pm 0.4$ |
| 14 | $36.3 \pm 0.5$ | $36.0 \pm 0.6$ | $\mathbf{37.0 \pm 0.6}$ | $\underline{36.4 \pm 0.5}$ | $31.4 \pm 0.8$ | $33.9 \pm 1.1$ | $33.7 \pm 0.2$ |
| 16 | $38.3 \pm 0.4$ | $38.2 \pm 0.6$ | $\mathbf{38.8 \pm 1.0}$ | $\underline{38.5 \pm 0.7}$ | $33.1 \pm 0.8$ | $35.9 \pm 0.6$ | $34.7 \pm 1.0$ |
| 18 | $40.1 \pm 0.3$ | $39.3 \pm 1.4$ | $\mathbf{40.7 \pm 0.5}$ | $\underline{40.7 \pm 0.5}$ | $34.8 \pm 0.9$ | $37.7 \pm 0.7$ | $36.9 \pm 0.5$ |
| 20 | $41.5 \pm 0.3$ | $40.9 \pm 0.9$ | $\underline{41.5 \pm 1.0}$ | $\mathbf{42.2 \pm 0.8}$ | $35.3 \pm 0.8$ | $39.2 \pm 0.9$ | $37.6 \pm 0.7$ |
| | | | | ImageNet-100 | | | |
| 1 | $20.4 \pm 1.1$ | $20.4 \pm 1.1$ | $20.4 \pm 1.1$ | $20.4 \pm 1.1$ | $\mathbf{25.9 \pm 1.1}$ | $20.4 \pm 1.1$ | $\underline{22.0 \pm 1.3}$ |
| 2 | $31.4 \pm 2.7$ | $30.7 \pm 1.3$ | $29.8 \pm 0.8$ | $30.0 \pm 1.2$ | $36.0 \pm 0.8$ | $\underline{36.5 \pm 0.8}$ | $\mathbf{36.8 \pm 0.9}$ |
| 4 | $45.5 \pm 2.2$ | $45.2 \pm 1.6$ | $45.7 \pm 1.1$ | $46.5 \pm 1.9$ | $\underline{47.1 \pm 1.9}$ | $46.6 \pm 1.3$ | $\mathbf{50.3 \pm 1.4}$ |
| 6 | $54.6 \pm 1.3$ | $56.9 \pm 2.0$ | $55.3 \pm 1.0$ | $\underline{57.3 \pm 1.4}$ | $55.2 \pm 0.7$ | $53.8 \pm 2.7$ | $\mathbf{58.5 \pm 0.5}$ |
| 8 | $60.1 \pm 1.4$ | $62.7 \pm 0.8$ | $61.2 \pm 1.6$ | $\mathbf{63.3 \pm 0.8}$ | $58.7 \pm 1.1$ | $58.8 \pm 4.2$ | $\underline{63.1 \pm 0.4}$ |
| 10 | $64.1 \pm 0.9$ | $\underline{66.6 \pm 1.9}$ | $66.4 \pm 0.9$ | $\mathbf{67.8 \pm 1.4}$ | $61.7 \pm 1.0$ | $63.2 \pm 4.0$ | $66.4 \pm 0.9$ |
| 12 | $67.3 \pm 0.5$ | $\underline{70.4 \pm 1.1}$ | $69.4 \pm 0.8$ | $\mathbf{70.5 \pm 1.2}$ | $64.3 \pm 1.0$ | $66.2 \pm 2.8$ | $68.8 \pm 0.9$ |
| 14 | $69.2 \pm 0.4$ | $\underline{72.4 \pm 0.9}$ | $71.9 \pm 0.4$ | $\mathbf{72.8 \pm 1.1}$ | $65.9 \pm 0.5$ | $69.0 \pm 1.8$ | $70.4 \pm 0.5$ |
| 16 | $71.3 \pm 0.6$ | $\underline{73.9 \pm 1.0}$ | $72.8 \pm 1.2$ | $\mathbf{74.4 \pm 0.8}$ | $67.1 \pm 0.5$ | $70.9 \pm 1.6$ | $72.0 \pm 0.4$ |
| 18 | $72.5 \pm 0.6$ | $\underline{75.1 \pm 0.8}$ | $74.5 \pm 0.7$ | $\mathbf{75.8 \pm 1.3}$ | $68.3 \pm 0.4$ | $73.0 \pm 1.3$ | $73.2 \pm 0.9$ |
| 20 | $73.5 \pm 0.9$ | $\underline{76.4 \pm 0.5}$ | $75.7 \pm 0.6$ | $\mathbf{77.2 \pm 0.9}$ | $68.7 \pm 0.8$ | $74.4 \pm 1.2$ | $74.3 \pm 0.6$ |
| | | | | DomainNet-Real | | | |
| 1 | $16.0 \pm 0.7$ | $16.0 \pm 0.7$ | $16.0 \pm 0.7$ | $16.0 \pm 0.7$ | $\mathbf{22.9 \pm 0.6}$ | $16.0 \pm 0.7$ | $\underline{17.9 \pm 0.6}$ |
| 2 | $25.2 \pm 0.6$ | $24.5 \pm 0.9$ | $25.8 \pm 0.7$ | $25.7 \pm 0.7$ | $29.2 \pm 0.7$ | $\underline{30.1 \pm 0.4}$ | $\mathbf{30.4 \pm 0.4}$ |
| 4 | $37.7 \pm 1.1$ | $38.3 \pm 0.6$ | $38.5 \pm 0.6$ | $38.9 \pm 0.9$ | $35.8 \pm 0.5$ | $\underline{39.2 \pm 0.5}$ | $\mathbf{41.9 \pm 0.2}$ |
| 6 | $45.6 \pm 0.6$ | $46.4 \pm 0.7$ | $46.4 \pm 1.0$ | $\underline{47.0 \pm 0.3}$ | $40.3 \pm 0.4$ | $43.9 \pm 0.5$ | $\mathbf{47.7 \pm 0.4}$ |
| 8 | $50.4 \pm 0.6$ | $51.5 \pm 0.3$ | $\underline{51.5 \pm 0.8}$ | $\mathbf{52.2 \pm 0.9}$ | $43.8 \pm 0.3$ | $46.9 \pm 0.5$ | $51.5 \pm 0.3$ |
| 10 | $54.1 \pm 0.4$ | $\underline{55.2 \pm 0.3}$ | $55.2 \pm 0.5$ | $\mathbf{55.8 \pm 0.4}$ | $46.5 \pm 0.6$ | $49.4 \pm 0.2$ | $54.3 \pm 0.3$ |
| 12 | $\mathbf{47.8 \pm 0.7}$ | $43.2 \pm 0.9$ | $\underline{45.9 \pm 1.8}$ | $43.9 \pm 1.2$ | $41.1 \pm 0.8$ | $43.9 \pm 1.7$ | $42.4 \pm 1.0$ |
| 14 | $57.6 \pm 0.2$ | $58.5 \pm 0.4$ | $\underline{58.8 \pm 0.1}$ | $\mathbf{59.3 \pm 0.1}$ | $50.2 \pm 0.6$ | $52.5 \pm 0.1$ | $56.8 \pm 0.3$ |
| 16 | $59.8 \pm 0.2$ | $60.6 \pm 0.3$ | $\underline{60.7 \pm 0.3}$ | $\mathbf{61.4 \pm 0.3}$ | $52.0 \pm 0.4$ | $54.0 \pm 0.3$ | $58.7 \pm 0.3$ |
| 18 | $61.2 \pm 0.1$ | $62.0 \pm 0.2$ | $\underline{62.2 \pm 0.3}$ | $\mathbf{62.8 \pm 0.2}$ | $53.9 \pm 0.3$ | $55.4 \pm 0.2$ | $60.2 \pm 0.1$ |
| 20 | $62.3 \pm 0.2$ | $\underline{63.4 \pm 0.2}$ | $63.4 \pm 0.3$ | $\mathbf{64.1 \pm 0.2}$ | $55.1 \pm 0.3$ | $56.7 \pm 0.2$ | $61.2 \pm 0.2$ |