# OpenReview forum: "Revisiting Active Learning in the Era of Vision Foundation Models"
_TMLR — Accepted by TMLR_

### Review · Reviewer_sEZP · 2024-02-19

**Summary Of Contributions:**

This paper mainly targets active learning. The authors try to explore the usage of foundation models in AL regarding several important component. Based on the finding they propose a new AL strategy which utilizes the inconsistency among predictions of input features applied with different dropout. The authors conduct several experiments to show the effectiveness of the proposed method.

**Audience:**

Yes

**Claims And Evidence:**

Yes

**Requested Changes:**

Please refer to the weaknesses.

**Strengths And Weaknesses:**

Strengths:
1. This paper provides meaningful experiment results explaining how foundation models can help in AL.

2. The proposed DropQuery is simple and effective.

Weaknesses:
1. It would be better to reorganize the paper so that more ablation study results can be presented in the main paper.

2. I wonder if such a method can be applied to supervised trained backbone models? Also, can such a method be applied to non-contrastive SSL backbone models, e.g. MAE?

3. Fig.2 can be improved for better understanding.

---

> ### Author Response · Authors · 2024-03-22
> **Response to Reviewer sEZP**
>
> We thank the reviewer for their feedback and are glad that they appreciate our extensive experimental results and the simplicity and effectiveness of our AL strategy.
>
> 1. **Ablation results:** We have conducted ablation results comparing the impact of the dropout ratio and $M$, the number of dropout iterations in Supplementary Section A.3. We agree with the reviewer that these results might be a better fit for the main text and will be revising it accordingly (see Section 5.4.2).
>
> 2. **Applicability to other backbones:** The DropQuery method proposed in the paper can be applied to any kind of backbone that maps an input image to an embedding vector, regardless of the backbone architecture or training strategy (supervised vs. self-supervised, contrastive vs. non-contrastive SSL). The key requirement for good AL performance in the low-budget regime is the quality of the embeddings generated by the backbone and the semantic structure of this representation space.
>
>    Currently, large vision transformers such as DINOv2 and OpenCLIP, which are trained using SSL methods on hundreds of millions, or even billions of images, are among the top choices for generating embeddings with rich semantic structure. As research in SSL evolves and new approaches are discovered, our method can still be applied with only minor modifications required to our modular code base to support newer backbones.
>
> 3. **Figure 2 changes:** In the interests of providing a comprehensive comparison of our method, we have benchmarked it against a multitude of other AL strategies. While the graphs may have many data points, we hope that it is clear that DropQuery generally outperforms other methods. A full breakdown of the numerical results is provided in Supplementary Section A.4. We would be happy to consider any specific changes to Figure 2 that improve clarity.

---

### Review · Reviewer_LkyM · 2024-03-01

**Summary Of Contributions:**

This work studies several design choices in Active Learning (AL) in the context of image classification based on frozen vision foundation models (e.g., DINO and CLIP), and proposes a new AL strategy, DropQuery. Through analytical experiments with DINOv2-ViT-g/14, various existing AL methods, and several image classification datasets, this work demonstrates that 1) the initial pool selection greatly influences the early stage of AL; 2) sample diversity and representativity should be emphasized in AL selection; and 3) utilization of semi-supervised learning may not be a necessary option. Based on such observations, this work proposes DropQuery, an effective AL strategy that 1) measures uncertainty of samples via dropout, 2) clusters high-priority samples, and 3) queries labels for samples closest to the cluster centroids. Experiments validate the efficacy of DropQuery on natural images and out-of-domain biomedical images.

**Audience:**

Yes

**Broader Impact Concerns:**

No significant adverse effects are apparent from this study. On the contrary, the proposed method enhances the training environment in data-scarce situations, thereby potentially broadening the accessibility of models for individuals who have restricted access to large-scale datasets.

**Claims And Evidence:**

Yes

**Requested Changes:**

Most of the requested changes have been illustrated above. In addition, these minor issues may be fixed:

1. In the paragraph right before Section 3.1, the query budget is set as “1 sample per class per iteration.” Using the word “class” may cause some confusion, because at the time of query, the class labels are unknown. According to later sections, “1 sample per cluster” may be more suitable.

2. The analytical experiments in Section 3 are all performed with DINOv2-ViT-g/14, which may lack generality. It may be helpful to show some more results considering other models and scales.

3. The evaluated datasets do not seem to involve long-tail distributions, which are actually common in real-world scenarios. It is unclear if the proposed clustering-based AL strategy is still helpful on long-tail classification tasks. Some attempts on long-tail classification or discussion may be beneficial.

4. Grammar Issue
    - [Abstract] “... efficiency, but …” → “... efficiency. But …”
    - [Introduction] “... where labeled data is scarce and …” → “... where labeled data are scarce and…”
    - [Sec 3.1] “The results some queries like…” is grammatically incorrect.
    - [Sec 3.4] “However, in our experiments find that” is grammatically incorrect.
    - Additional errors throughout the manuscript necessitate thorough proofreading and correction.

**Strengths And Weaknesses:**

Strengths
1. This work studies AL for image classification based on pre-trained large-scale vision/vision-language models. This study can bring insights for applications of vision foundation models in domain-specific, low-data tasks.

2. The evaluated datasets are comprehensive, including both natural images and out-of-domain (medical) images.

3. The comparison between the proposed AL strategy DropQuery and prior methods clearly shows the advantages of DropQuery.

4. The writing is generally clear and easy to follow.

Weaknesses
1. [Impact of initial pool (in the long run)] In Section 3.1, AL methods starting with a randomly sampled initial pool are compared with those starting from a centroid-based initial pool, and the results “dispel previous preliminary findings Chandra et al. (2020) that emphasize that there is no significant difference in initial pool selection strategies for AL.” However, from the reviewer’s perspective, the conclusions of this work and Chandra et al. (2020) do not seem to contradict each other. In this work, Table 1 indeed demonstrates that the centroid-based initial pool leads to greatly improved performance at the very beginning of AL (e.g., t=1). However, as the number of iterations increases, the benefits diminish or even become negative (e.g., t=8 or 16 in ImageNet-100). The observation from Chandra et al. (2020) is that “Experimental results could not conclusively prove that intelligently sampled initial pools are better for AL than random initial pools *in the long run*,” which do not seem to be very different from the results in Section 3.1. It is suggested to provide some clarification or revise the related claims to avoid misunderstandings of the conclusions in this work.

2. [Utilization of unlabeled instances] In Section 3.4, it is claimed that “the efficacy of *semi-supervised learning* in this setting is questionable at best, with wide variation across query methods and datasets,” based on experiments of (an offline variant of) label propagation (Iscen et al. (2019)). However, this method of utilizing unlabeled instances is just one example of semi-supervised learning, and the conclusion for ineffective semi-supervised learning may not generalize well to other methods, such as Mean Teacher (Tarvainen and Valpola (2017)), self-training (Yalniz et al. (2019)), or Meta Pseudo Labels (Pham et al. (2021)).

3. [Transition from observations to proposal] The connection between the analytical experiments (Section 3) and the proposed new AL strategy (Section 4) seems not very strong. It may be helpful to directly point out which part of DropQuery originates from the observations.

Akshay L Chandra, Sai Vikas Desai, Chaitanya Devaguptapu, and Vineeth N. Balasubramanian. On initial
pools for deep active learning. In Preregister@NeurIPS, 2020.

Antti Tarvainen and Harri Valpola. Mean teachers are better role models: Weight-averaged consistency targets improve semi-supervised deep learning results. In NeurIPS, 2017.

I. Zeki Yalniz, Hervé Jégou, Kan Chen, Manohar Paluri, Dhruv Mahajan. Billion-scale semi-supervised learning for image classification. arXiv, 2019.

Hieu Pham, Zihang Dai, Qizhe Xie, Minh-Thang Luong, Quoc V. Le. Meta Pseudo Labels. In CVPR, 2021.

---

> ### Author Response · Authors · 2024-03-22
> **Response to Reviewer LkyM**
>
> We thank the reviewer for their detailed feedback and are glad that they appreciated our comprehensive experiments and the efficacy of our proposed method.
>
> 1. **Impact of initial pool:** We acknowledge that our claims could be improved with additional nuance to avoid misunderstandings. While Chandra et al. didn't find any evidence of improved performance in the long run, we would like to draw the reviewer's attention to the results of their main experiments (Fig 2-5) in which the accuracy curves for all methods but VAE almost perfectly overlap with one another, not just in the long run (60% labels) but also in the initial iterations (10% labels). We observed a significant improvement in the initial iterations of our experiments and it is precisely this discrepancy that we wish to highlight.
>
>    If a researcher considering an AL strategy were to base their approach solely on the findings of Chandra et al., they might conclude that initial pool selection has little to no impact in the short run. That being said, we agree with the reviewer that our conclusions are not contradictory because of the differences in the labeling budgets in our respective studies. As you correctly point out, the benefits of intelligent initial pool selection seem to diminish with later iterations. However, the number of data points in our experimental iterations are still much less than the starting budget for the experiments in Chandra et al., which don't explore as low of a budget as our experiments. Given that this nuance was lacking from our initial submission, we will include it in the revised version (see the last few paragraphs of Section 3.1).
>
> 2. **Utilization of unlabeled instances:** We agree with the reviewer's observation that label propagation is just one method of semi-supervised learning (SSL), and other SSL methods might perform better. Due to our experimental setup involving a fixed representation space generated by frozen foundation model backbones, we are constrained to a subset of SSL algorithms that operate in an offline mode, unlike SOTA methods like FlexMatch which was also employed by the TypiClust strategy.
>
>    Typically, most research into SSL shows some improvement in performance by using unlabeled instances, with better methods demonstrating greater improvements. It is not usually the case that performance decreases, or as we demonstrate in the case of AL with foundation model features, performance gains are questionable, and we believe that this result is worth highlighting. We acknowledge that this warrants further investigation with other methods including those referenced by the reviewer, but would defer those experiments to future work. Within the limited scope of this paper, we will clarify that our recommendation is limited to the use of the label propagation algorithm and may not be representative of other SSL methods.
>
> 3. **Transitions from observation to proposal:** The key elements of DropQuery -using the centroid-based initial pool selection to overcome the cold-start problem, prioritizing uncertain samples over representative samples, and selecting diverse points via candidate clustering- are all chosen based on the objective analysis presented in Section 3. We acknowledge that this connection could be made more explicit and will revise our writing accordingly (see Section 3.4 / 4).
>
> 4. **Model scales:** The goal of our experiments was to highlight how rich representation spaces generated by large-scale foundation models influence various factors of Active Learning strategies. Given this consideration, we decided to pick the largest variant of DINOv2 for our investigations in Section 3. As foundation models improve over time, we wished to highlight the best-case scenario in contrast to exploring how model scale influences AL methods.
>
>    Our limited probing into model scales in Figure 1. demonstrates that the largest foundation models exhibit the most drastic improvements in performance. While we acknowledge that more experiments into model scales would provide interesting insights, from a practitioner's perspective, there are limited reasons to not use giant VIT backbones, the most compelling being computational constraints. However, we were able to generate embeddings with these models for our datasets with a single GPU in a few minutes.
>
> 5. **Long-tailed datasets:** We agree that exploring long-tailed datasets would be an interesting direction of research, particularly in the biomedical imaging domain where such scenarios are the norm. Our datasets for experimentation were chosen based on well-known benchmarks, and we wished to scale them up to millions of images (Places365) to push for more realistic applications since many existing AL strategies only test their methods on smaller-scale datasets like MNIST or CIFAR10/100. We will include a discussion on long-tailed datasets and highlight them as an exciting direction for further study (see Limitations).

---

### Review · Reviewer_Zp5y · 2024-03-08

**Summary Of Contributions:**

This paper systematically studies four critical components of effective active learning in the context of Foundation Models. Based on the results of these researches, this paper challenges established research on the cold-start problem and counter the notion that uncertainty-based queries are ineffective in the low-budget regime. Finally, this paper introduces a new AL strategy called DropQuery.

**Audience:**

Yes

**Broader Impact Concerns:**

No.

**Claims And Evidence:**

Yes

**Requested Changes:**

Please kindly refer to the above comments.

**Strengths And Weaknesses:**

**Strengths**

1.	This paper carries out deep researches on critical components of active learning.

2.	The advantages of the proposed strategy are verified from several angles.

**Weaknesses**

1.	My main concern is that the main idea of this article is not clear. After reading the contributions of this article, I think the main contribution is the new strategy. But in fact, not only the title, but also the main part of the article is about effects of the four critical components of effective active learning. I think authors should be clear of what they want to show us mostly and modify the structure of this article.

2.	In section 3.2, this paper carries out a large number of experiments and shows us the results. However, this paper only gives experiments settings and lacks the analysis of the experimental results. This is also the case in sections 3.1, 3.3, and 3.4.

3.	The findings of the experiment should be clearly stated.

4.	The setting of parameter M in Section 4 is not clear. It should be explained why M=3 not something else.

5.	There is a lack of comparison with baselines in the experiments in Section 5.2. The results can only tell us that DrouQuery works on biomedical datasets. But is it really better than other existing strategies?

---

> ### Author Response · Authors · 2024-03-22
> **Response to Reviewer Zp5y**
>
> We thank the reviewer for their constructive feedback and are glad that they appreciate the depth of our research exploring Active Learning with Vision Foundation Models.
>
> 1. **Main contributions:** You are correct that the DropQuery AL strategy is one of our key contributions. However, we emphasize that the creation of DropQuery follows from our other key contribution - a systematic investigation of 4 components of AL strategies in the context of vision foundation models. Given the long history of AL, many previous works leveraged the models and datasets available at the time. Our work first seeks to understand how these previous findings are influenced by the usage of foundation models, a relatively recent contribution.
>
>    The results of this detailed investigation of 4 components are themselves important contributions to future research in AL, and these insights can be used to design better and more performant AL strategies that take advantage of the benefits offered by foundation models. To prove the value of these insights, we design a simple new strategy, DropQuery, which is constructed by directly incorporating the findings of our experiments in Section 3, and the comparisons with other AL methods in Section 5 demonstrate its superior performance.
>
>    We have outlined our contributions in the Introduction, but will revise that section to highlight our contributions of 1) more clearly our experimental insights, and 2) a new AL strategy informed by these insights.
>
> 2. **Analysis of experimental results:** In Section 3.2, we highlight the importance of querying diverse samples that span the representation space of the dataset. We demonstrate this by showing that modifying simple uncertainty based AL query strategies (Uncertainty, Entropy, Margins, BALD) to perform more diverse sampling enables them to match or surpass more sophisticated strategies which are explicitly designed to sample diverse points. Since we keep all other variables constant, the addition of diversity sampling to the simpler queries is the decisive factor underlying the significant improvements in performance over the baseline implementation.
>
> 3. We will revise our writing to clarify this result and the other results in Section 3 by adding a new section (see Section 3.5) summarizing the analysis and conclusions from our investigation of the critical components of AL.
>
> 4. We have explored the influence of the hyperparameter $M$ in Supplementary Section A.3.2. The strategy is robust to the choice of $M$ with no benefit to selecting larger values. Since the time taken for the creation of the candidate set $Z_c$ is $O(M)$, we select the smallest value of $M=3$, which enables efficient scaling to larger datasets without compromising on performance. Since this point has been highlighted in other reviews as well, we will move this discussion to the main text to clarify our choice of $M$ (see Section 5.4.2).
>
> 5. The results in Section 5.2, focusing on biomedical imaging, are complemented by results in Section 5.1 on 4 natural image datasets (Cars, Aircraft, Pets, Places365) of varying sizes and complexities. Additionally, we include full numerical results on the previous 4 natural image datasets used in Section 3 (CIFAR100, Food101, ImageNet-100, DomainNet-Real) and 2 more biomedical datasets (colorectal histology and Patch Camelyon) in the supplementary section A.4. for a total of 14 datasets which clearly demonstrates DropQuery's performance across a large spectrum of settings.

---

### Decision · Action_Editor_PW4P · 2024-05-19

**Recommendation:** Accept with minor revision

**Comment:**

The paper explores the integration of vision foundation models into active learning (AL) strategies, particularly focusing on low-budget scenarios. The authors evaluate the impact of using foundation models on three crucial aspects of AL: initial labeled pool selection, diversity sampling, and the balance between representativeness and uncertainty in sampling. Their findings inform a newly developed AL strategy, "DropQuery," which combines uncertainty estimation through dropout with sample diversity to enhance AL efficiency. This strategy is extensively tested across various challenging image classification benchmarks, including natural and biomedical images, demonstrating its potential to refine active learning processes in complex image datasets.

The paper is overall solid. The authors thoroughly investigate critical components of effective AL, offering new insights particularly useful for domain-specific, low-data tasks. The investigation of AL with the context of foundation models is interesting and valuable. The proposed AL strategy, DropQuery is simple yet effective across various tests. On the weakness side, several reviewers noted that the main ideas and contributions could be clearer and more distinctively articulated within the paper. The authors were able to address some concerns with their rebuttal, but the paper could further benefit from appropriate rewriting and re-organization. The authors should better address reviewer sEZP's questions with the following changes:
1. Add the MAE backbone experiment besides DINOv2.
2. Re-organize and improve the paper according to the suggestions.

**Audience:**

Yes. The findings of this paper would be of interest to the general (multimodal) foundation model, active learning, and transfer learning communities. The investigation of different AL aspects and its combination with foundation model are relevant and valuable to the field.

**Claims And Evidence:**

The claims made in the submission are generally supported by evidence. The experimental setup and the results are comprehensive. The proposed DropQuery strategy is effective for active learning with foundation models.

---

> ### Author Response · Authors · 2024-06-18
> **Response to Action Editor**
>
> We thank the action editor and reviewers for taking the time to provide us with valuable feedback to improve our manuscript. We are pleased that it has been recommended for acceptance with minor revisions.
>
> In response to the question about MAE backbones from reviewer sEZP, we have conducted additional experiments and included their results in the supplementary material. Our findings are briefly summarized as follows:
>
> - We tested the ViT-H/14 MAE pretrained model introduced in [1] with 7 AL strategies on 4 datasets under the same experimental settings as previous experiments.
> - We observed inconsistent results with many queries being similar in performance to random sampling.
> - Methods relying on clustering features were typically outperformed by random sampling, whereas those which did not leverage the structure of the representation space performed better.
> - ImageNet-1K pretrained MAE is not a good feature extractor for other datasets. It also doesn't seem to be well suited to the very low-budget regime we study.
> - The difference in trends between the MAE backbone and the DINOv2 models (which also include a masked image modelling loss term) highlights the need to revisit AL in the context of foundation models, so that strategies can be developed to take full advantage of the rich representation spaces of these models.
>
> [1] https://arxiv.org/abs/2111.06377